EMBO
Molecular Medicine

# Cholesterol-loaded nanoparticles ameliorate synaptic and cognitive function in Huntington's disease mice

Marta Valenza[1,†], Jane Y Chen[2,†], Eleonora Di Paolo[1,‡], Barbara Ruozi[3,‡], Daniela Belletti[3], Costanza Ferrari Bardile[1], Valerio Leoni[4,5], Claudio Caccia[4], Elisa Brilli[1], Stefano Di Donato[4,§], Marina M Boido[6], Alessandro Vercelli[6], Maria A Vandelli[3], Flavio Forni[3], Carlos Cepeda[2], Michael S Levine[2], Giovanni Tosi[3] & Elena Cattaneo[1,*]

## Abstract

Brain cholesterol biosynthesis and cholesterol levels are reduced in mouse models of Huntington's disease (HD), suggesting that locally synthesized, newly formed cholesterol is less available to neurons. This may be detrimental for neuronal function, especially given that locally synthesized cholesterol is implicated in synapse integrity and remodeling. Here, we used biodegradable and biocompatible polymeric nanoparticles (NPs) modified with glycopeptides (g7) and loaded with cholesterol (g7-NPs-Chol), which *per se* is not blood–brain barrier (BBB) permeable, to obtain high-rate cholesterol delivery into the brain after intraperitoneal injection in HD mice. We report that g7-NPs, in contrast to unmodified NPs, efficiently crossed the BBB and localized in glial and neuronal cells in different brain regions. We also found that repeated systemic delivery of g7-NPs-Chol rescued synaptic and cognitive dysfunction and partially improved global activity in HD mice. These results demonstrate that cholesterol supplementation to the HD brain reverses functional alterations associated with HD and highlight the potential of this new drug-administration route to the diseased brain.

**Keywords** cholesterol; cognition; Huntington's disease; nanoparticles; synapses

**Subject Categories** Metabolism; Neuroscience

## Introduction

Huntington's disease (HD) is a genetic neurological disorder caused by a CAG expansion in the gene encoding the huntingtin (HTT) protein (HDCRG, 1993). Clinically, HD is characterized by motor, cognitive, and psychiatric disturbances (Ross *et al*, 2014) and is associated with neuronal dysfunction, atrophy of the striatum and other brain regions, and progressive loss of striatal medium-sized spiny neurons (MSNs) and of cortical pyramidal neurons (Vonsattel & DiFiglia, 1998). Several molecular and cellular dysfunctions have been identified (Zuccato *et al*, 2010), and one affected pathway implicates brain cholesterol.

The brain is the most cholesterol-rich organ in the body, with almost all of the cholesterol produced *in situ*, as circulating cholesterol is not able to cross the BBB (Dietschy & Turley, 2004). A large majority of cholesterol (> 70% of brain cholesterol mass) is present in myelin sheaths. Indeed, the rate of cholesterol synthesis is highest during post-natal stage to build myelin scaffolding. Cholesterol is also a structural component of glial and neuronal membranes and is concentrated in lipid rafts, specialized membrane microdomains that initiate, propagate, and maintain signal transduction events (Paratcha & Ibanez, 2002). Newly synthesized cholesterol is also required for vesicle assembly and fusion (Huttner & Zimmerberg, 2001; Lang *et al*, 2001), synapse formation, integrity, remodeling (Pfrieger, 2003), and neurotransmitter release (Thiele *et al*, 2000; Mauch *et al*, 2001). Accordingly, a breakdown of cholesterol synthesis causes brain malformations and impaired cognitive functions (Valenza & Cattaneo, 2006).

1 Department of BioSciences, Centre for Stem Cell Research, Università degli Studi di Milano, Milan, Italy
2 Intellectual and Developmental Disabilities Research Center, Semel Institute for Neuroscience, Brain Research Institute, David Geffen School of Medicine, University of California Los Angeles, Los Angeles, CA, USA
3 Department of Life Sciences, University of Modena and Reggio Emilia, Modena, Italy
4 Neurological Institute C. Besta, Milan, Italy
5 Laboratory of Clinical Chemistry, Ospedale di Circolo e Fondazione Macchi, Varese, Italy
6 Neuroscience Institute Cavalieri Ottolenghi, Neuroscience Institute of Turin, Orbassano, Turin, Italy
*Corresponding author. Tel: +39 02 50325842; E-mail: elena.cattaneo@unimi.it
†These authors share first authorship
‡These authors share second authorship
§Deceased on 12 November 2015

HD is characterized by abnormal brain cholesterol homeostasis. Patients with HD show altered cholesterol homeostasis since pre- and early stages of disease as judged by the plasmatic measure of 24S-hydroxy-cholesterol (24OHC), the brain-specific catabolite of cholesterol able to cross the blood–brain barrier (BBB) (Leoni *et al*, 2008, 2013). Reduced cholesterol biosynthesis and levels are also found in the brain of several HD mouse models (Valenza *et al*, 2007a,b, 2010). On the contrary, others reported an increased accumulation of free cholesterol in brain tissues of HD mouse models (Trushina *et al*, 2006; del Toro *et al*, 2010) likely due to different sample preparation and less sensitive methods (colorimetric and enzymatic assays) to detect and quantify cholesterol compared to mass spectrometry (Marullo *et al*, 2012). Of note, more recently, some of the same groups have reported a decrease of lathosterol and cholesterol levels in the striatum of a HD mouse model by means of mass spectrometry (Trushina *et al*, 2014). Cholesterol dysregulation occurs in astrocytes (Valenza *et al*, 2015) and is linked to a specific action of mutant HTT on sterol regulatory-element-binding proteins (SREBPs) and its target genes, whose reduced transcription leads to less brain cholesterol produced and released and available to be uptaken by neurons (Valenza *et al*, 2005).

Accordingly, an early decrease of cholesterol production in the HD brain might be detrimental for neuronal activities. Abnormalities in synaptic communication within the striatum and between the cortex and striatum occur long before, or in the absence of, cell death in HD animal models (Milnerwood & Raymond, 2010) and cognitive disturbances have been observed decades before predicted clinical diagnosis in HD gene carriers (Levine *et al*, 2004; Paulsen & Long, 2014). Similarly, brain cholesterol biosynthesis is significantly reduced before the onset of motor symptoms in all the HD animal models analyzed so far (Valenza *et al*, 2007a,b) and synaptosomes —a compartment dedicated to impulse transmission and neurotransmitter release—carry suboptimal levels of sterols in the early stages of HD in one mouse model (Valenza *et al*, 2010). However, a link between the reduced level of cholesterol and neuronal dysfunction *in vivo* in HD is still missing.

Here, we explored the effects of cholesterol supplementation on synaptic communication and machinery, motor and cognitive behaviors, and neuropathology in the R6/2 mouse model, a well-established early onset transgenic mouse model of HD (Mangiarini *et al*, 1996). Since cholesterol does not cross the BBB, cholesterol was delivered using a new technology for drug administration in the brain (Vergoni *et al*, 2009; Tosi *et al*, 2010), that is, via biodegradable polymeric (polylactide-co-glycolide, PLGA) nanoparticles (NPs) modified with a glycopeptide (g-7) able to cross the BBB upon systemic injection in mice (Costantino *et al*, 2005; Tosi *et al*, 2007, 2011b). The development of new strategies to enhance brain delivery based on colloidal carriers is of great importance, since nanocarriers can protect drugs and deliver them across the BBB to target brain cells in a non-invasive way (Tosi *et al*, 2008). Notably, both FDA and EMA have approved PLGA in various drug delivery systems in humans (Mundargi *et al*, 2008), as confirmed by a number of market products (i.e., Lupron Depot®, Nutropin Depot ®).

We report that, in contrast to unmodified NPs, g7-NPs efficiently crossed the BBB and within a few hours after systemic injection reached glial and neuronal cells in different brain regions.

Importantly, repeated systemic delivery of g7-NPs-Chol rescued synaptic communication, protected from cognitive decline and partially improved global activity in HD mice.

# Results

## Chemical–physical and technological optimization of unloaded and cholesterol-loaded Nanoparticles

The chemical formulation and features of unloaded NPs (u-NPs) herein employed have been largely described (Vergoni *et al*, 2009; Tosi *et al*, 2011a, 2014; Vilella *et al*, 2014). To optimize the production of NPs loaded with cholesterol (NPs-Chol), we first prepared u-NPs and NPs loaded with different amounts of cholesterol (1, 5, and 10 mg of Chol per 100 mg of polymer; herein defined as NPs-Chol1, NPs-Chol2 and NPs-Chol3, respectively) according to the nanoprecipitation procedure (Minost *et al*, 2012) (see Materials and Methods). The composition of different NPs is described in Appendix Table S1, and details about their optimization and characterization are described in the Appendix.

NPs were characterized by their chemical–physical properties, summarized in Appendix Table S2. The average diameter (Z-average) of u-NPs ranged from 170 to 192 nm. Z-average for NPs-Chol1 and NPs-Chol2 was lower than 210 nm, while size of NPs-Chol3 ranged between 200 nm and 300 nm. The polydispersity index (PDI value), a measure of the heterogeneity of NPs, was $0.08 \pm 0.01$ for u-NPs, suggesting a homogeneous and monomodal distribution population around the mean size. NPs-Chol1 and NPs-Chol2 showed a PDI value of $0.09 \pm 0.01$ and $0.11 \pm 0.02$, respectively, and a narrow dimension distribution, indicating that they are monomodal and monodisperse systems. On the contrary, NPs-Chol3 was characterized by a PDI value close to 0.3, accounting for a marked increase in sample heterogeneity. Zeta-potential (ζ-pot), a function of particle surface charges that influences cell interaction, was negative for all the NPs-Chol samples and similar to those of u-NPs. Moreover, ζ-pot of NPs-Chol3 displayed higher standard deviation $(-12 \pm 10$ mV$)$ with respect to those of NPs-Chol1 $(-9 \pm 4$ mV$)$ and NPs-Chol2 $(-8 \pm 4$ mV$)$, further highlighting the higher heterogeneity of this sample.

To evaluate whether and how the incorporation of cholesterol influences the morphology, architecture and surface properties of NPs, atomic force microscopy (AFM) and transmission electron microscopy (TEM) analyses were performed on u-NPs and NPs-Chol (Fig 1A–C). In agreement with the chemical–physical properties (Appendix Table S2), the "height" AFM image (Fig 1A, left column), 3D reconstruction (Fig 1A, middle column), and TEM micrograph (Fig 1A, right column) of u-NPs highlighted well compact and defined spherical structures (Belletti *et al*, 2012). The AFM analysis for NPs-Chol1 confirmed the spherical shape, but shape and size were less homogeneous if compared with those of u-NPs (Fig 1B). Particles adopted an irregular frame, evident in the AFM 3D reconstruction, supporting the hypothesis that alteration of polymer organization and intimate interplay between cholesterol and PLGA occurred when cholesterol was added to the formulation. The greater complexity of these samples was confirmed by TEM microphotographs (right columns) emphasizing the less dense and compact structures of NPs-Chol1 with respect to

                                    

u-NPs. NPs-Chol2 showed similar morphology and architecture of NPs-Chol1 (data not shown). Instead, the AFM images of NPs-Chol3 showed the presence of irregular structures and unformed material and a remarkable tendency to aggregate (Fig 1C). With respect to u-NPs and NPs-Chol1, NPs-Chol3 seemed to promote the formation of disorganized clusters characterized by heterogeneous dimensions (242 ± 52 nm) and by a roughness surface with evident fissuring. Similarly, TEM microphotographs showed the complexity of NPs-Chol3 that appeared with abundant adsorbed unformed material (likely unloaded cholesterol) and modified NPs' morphology.

We also evaluated the content of cholesterol into NPs (loading capacity, LC%) and the encapsulation efficiency (EE%) (Appendix Table S2). About 0.7 ± 0.1 mg/100 mg of formulation, corresponding to an EE of 68%, were loaded in the NPs-Chol1, indicating that an important fraction of the initial cholesterol was stably incorporated into the NPs-Chol1. On the contrary, a decrease in EE value was observed as the amount of cholesterol used in the preparation increased. In NPs-Chol2 and NPs-Chol3, the EE remarkably decreased (about 20%) although the highest value of drug loading was observed in NPs-Chol3 (2.5 mg of Chol/100 mg of NPs). However, as previously pointed out, cholesterol in NPs-Chol3 was

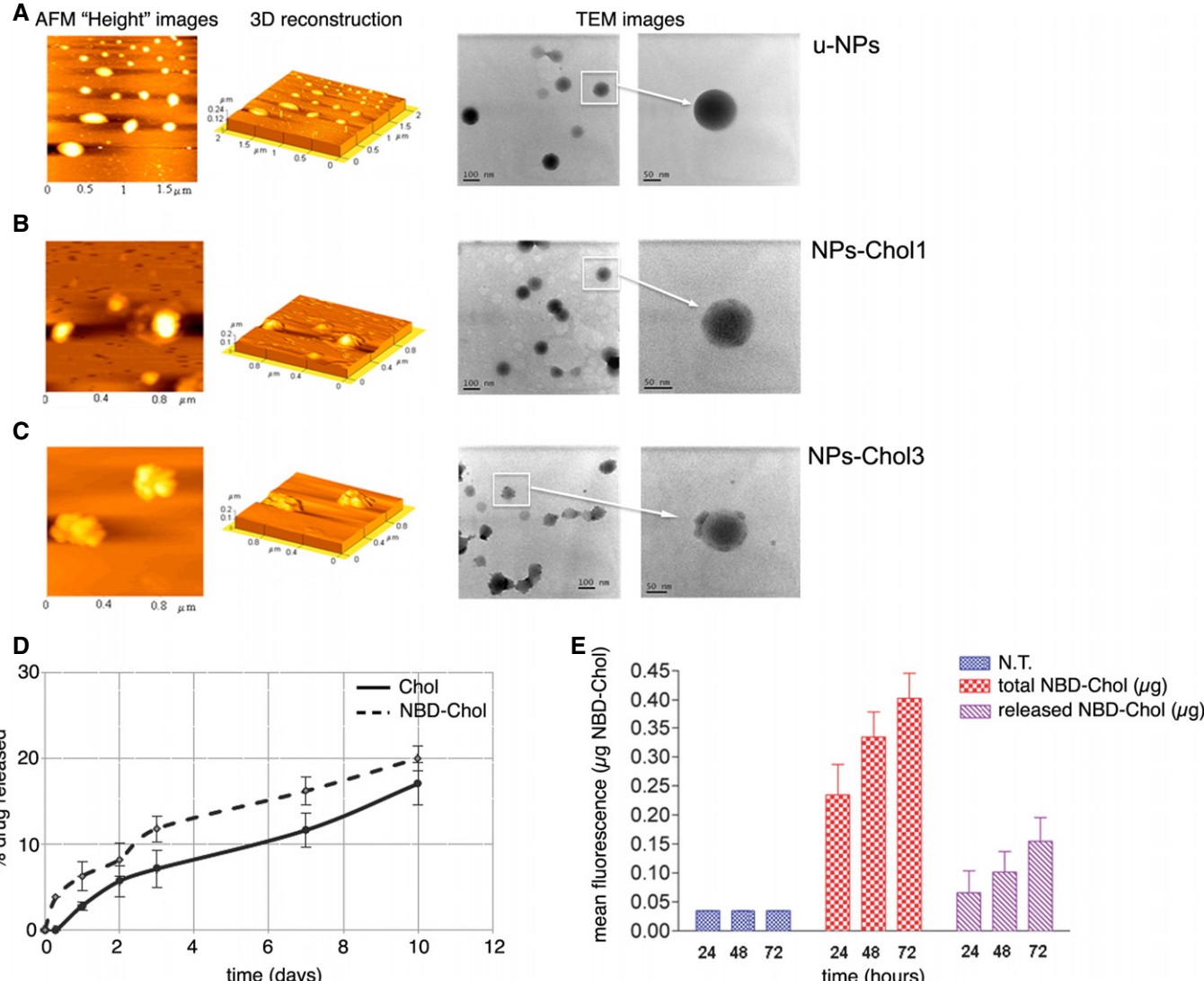

**Figure 1.  Characterization of NPs loaded with different concentrations of cholesterol.**

A–C   AFM and TEM analysis of unloaded (u-NPs) and cholesterol-loaded NPs (NPs-Chol). AFM "height" images (left column), 3D reconstruction (middle column), and TEM micrograph (right column) of u-NPs (A), NPs-Chol1 (B), and NPs-Chol3 (C).

D   Release profile in water of cholesterol (continuous line, —) and NBD-Chol (dotted line, - - -) from NPs-Chol1 and NPs-NBD-Chol1, respectively. The graph represents mean ± SEM. Data are from three independent experiments.

E   *In vitro* release of NBD-Chol from NPs at different time intervals in NS cells. Data in the graph represent mean (μg) ± SEM of total NBD-Chol (embedded into and released from NPs; red columns) and NBD-Chol released after NPs degradation (purple columns) present in the homogenates of NS cells treated with NPs-NBD-Chol1. Data obtained from four independent experiments. N.T.: not treated cells.

not completely embedded, but a remarkable fraction was absorbed onto the surface. Based on these analyses, NPs-Chol1 formulation was used in all experiments.

### Controlled release of cholesterol from NPs in physiological conditions and *in vitro*

To explore the ability of the system to release cholesterol, we first carried out release studies in deionized water for 10 days (Fig 1D). The release profile of cholesterol from NPs-Chol1 (hereafter referred as NPs-Chol; solid line) showed an initial "burst release" (< 8%) during the first 3 days, followed by a second slow release phase. Chol release was detected close to values of 18% over 10 days owing to the poor water solubility of cholesterol (estimated to be 2 µg/ml). Moreover, during the second phase, the slow linear release kinetic of Chol from NPs-Chol between day 5 and day 10 could be ascribed to NPs degradation.

In specific experiments, we also adopted a lead formulation prepared by replacing cholesterol with the fluorescent cholesterol derivative NBD-Chol to discriminate between endogenous and exogenous cholesterol released from NPs. We therefore characterized also the NBD-Chol-loaded NPs (NPs-NBD-Chol) in terms of their chemical–physical and technological properties (Appendix Table S2) and morphological features (Appendix Fig S1). The release of NBD-Chol from NPs in water showed a slow kinetic profile (Fig 1D, dotted line) similar to that observed for native cholesterol (Fig 1D, solid line). Similar findings were observed when the kinetic profile of drug release was evaluated in experiments conducted in cultured cells (Fig 1E). Spectrophotometric quantification of NBD-Chol in neural stem (NS) cells treated with 3 µg of NPs-NBD-Chol revealed that only 20% of the total NBD-Chol taken by the cells was released after 24 h (0.05 µg vs. 0.23 µg; Fig 1E, seventh and fourth columns, respectively). At 72 h, the amount of NBD-Chol released increased to about 35% of the total NBD-Chol taken up by cells (0.14 µg vs. 0.39 µg; Fig 1E, ninth and sixth columns, respectively), confirming the slow kinetic profile of cholesterol release from NPs.

### g7-NPs distribution in HD cells and brain

The g7-NPs used in this study are designed to cross the BBB, and previous studies indicated that about 10% are estimated to penetrate the brain (Costantino *et al*, 2005; Tosi *et al*, 2007, 2011a,b, 2014). To verify that g7-NPs could penetrate HD cells, primary neurons from R6/2 mice and neurons and astrocytes from mouse NS cells carrying 140 CAG repeats (NS Q140/7) were exposed to g7-NPs labeled with rhodamine to allow their detection with fluorescence microscopy. Appendix Fig S2 shows that g7-NPs are taken up *in vitro* by different brain cells expressing mutant Htt. Importantly, 4 h after a single intraperitoneal (ip) injection into 8-week-old R6/2 mice and wild-type (WT) littermates, both control (unmodified) NPs (C-NPs) and g7-NPs were detected in the liver (Fig 2A) and in other peripheral tissues (Appendix Fig S3), but only g7-NPs were detected in the brain (Fig 2B). Quantification of g7-NPs yielded an approximate ratio of ~10:1 in the WT liver compared to striatum and cortex (Fig 2C). This quantification also revealed a reduced propensity of g7-NPs to reach the R6/2 brain compared to the WT brain,

while g7-NPs were more prevalent in R6/2 liver compared to WT liver, suggesting that HD-related mechanisms may influence the BBB crossing of g7-NPs. g7-NPs were also found 24 h and 2 weeks after a single (Fig 2D) or multiple ip injections performed in the same week (Fig 2E). High-magnification confocal images indicated the presence of g7-NPs in different brain regions and in IBA1 immunoreactive microglial cells (Fig 2F) and in GFAP positive astrocytes (Fig 2G). Notably, g7-NPs were also detected in neuronal cells, as demonstrated by immunostaining against calbindin (Fig 2H; Appendix Fig S4) and DARPP-32 (Fig 2I).

### Delivery and release of cholesterol *in vivo* in the R6/2 brain

To track the delivery and intracellular release of cholesterol from g7-NPs, we employed rhodamine-labeled g7-NPs (Vergoni *et al*, 2009) loaded with the fluorescent cholesterol derivative NBD-Chol (g7-NPs-NBD-Chol). NBD-Chol closely resembles the structure of native cholesterol and is normally used to study cholesterol trafficking (Gimpl & Gehrig-Burger, 2007). Accordingly, NBD-Chol, injected into brain ventricles of mice, co-localizes with PMCA ATPase, a marker of plasma membrane, suggesting that exogenous cholesterol is incorporated on brain cells' membranes *in vivo* (Appendix Fig S5). We next monitored the distribution of g7-NPs as red spots and the distribution of released NBD-Chol as green signal. *In vivo*, at 12 and 24 h after a single ip injection of g7-NPs-NBD-Chol, g7-NPs and NBD-Chol co-localized in brain cells (Fig 3A and B). In particular, Fig 3B shows the distribution of g7-NPs (red signal) and NBD-Chol (green signal) in a brain section of a R6/2 mouse injected ip with g7-NPs-NBD-Chol and sacrificed 24 h later. Both g7-NPs and NBD-Chol signals co-localized as indicated by the scatterplot of red and green pixel intensities. However, g7-NPs and NBD-Chol were no longer co-localized after 14 days as demonstrated in Fig 3C. Similar results were found at 7 days after ip injection (data not shown). These findings indicate that NBD-Chol was partially released from NPs 1–2 weeks after injection, in parallel with a reduction in the signal from g7-NPs, probably due to their degradation. Quantification of g7-NPs in brain slices from injected mice confirmed a decreased number of NPs over time as determined after normalizing the red spots on the mean size of NPs (Fig 3D). In the liver, the kinetics of NBD-Chol release and g7-NPs degradation was faster (< 24 h) than in brain (Appendix Fig S6).

### g7-NPs-Chol rescue synaptic activity in HD mice

As synaptic transmission in striatal MSNs is altered in R6/2 mice during disease progression (Cepeda *et al*, 2003, 2004), we next explored whether cholesterol supplementation to the brain via systemic injection of g7-NPs-Chol restored synaptic parameters in HD mice. Pilot experiments with R6/2 animals that received only 1 or 2 injections of g7-NPs-Chol did not show any significant modifications in electrophysiological properties (data not shown). We therefore designed our trials in order to provide sustained and prolonged delivery of cholesterol to the HD brain. Treatment started at the pre-symptomatic stage (5 weeks of age) and continued until the symptomatic stage (9 weeks of age) under

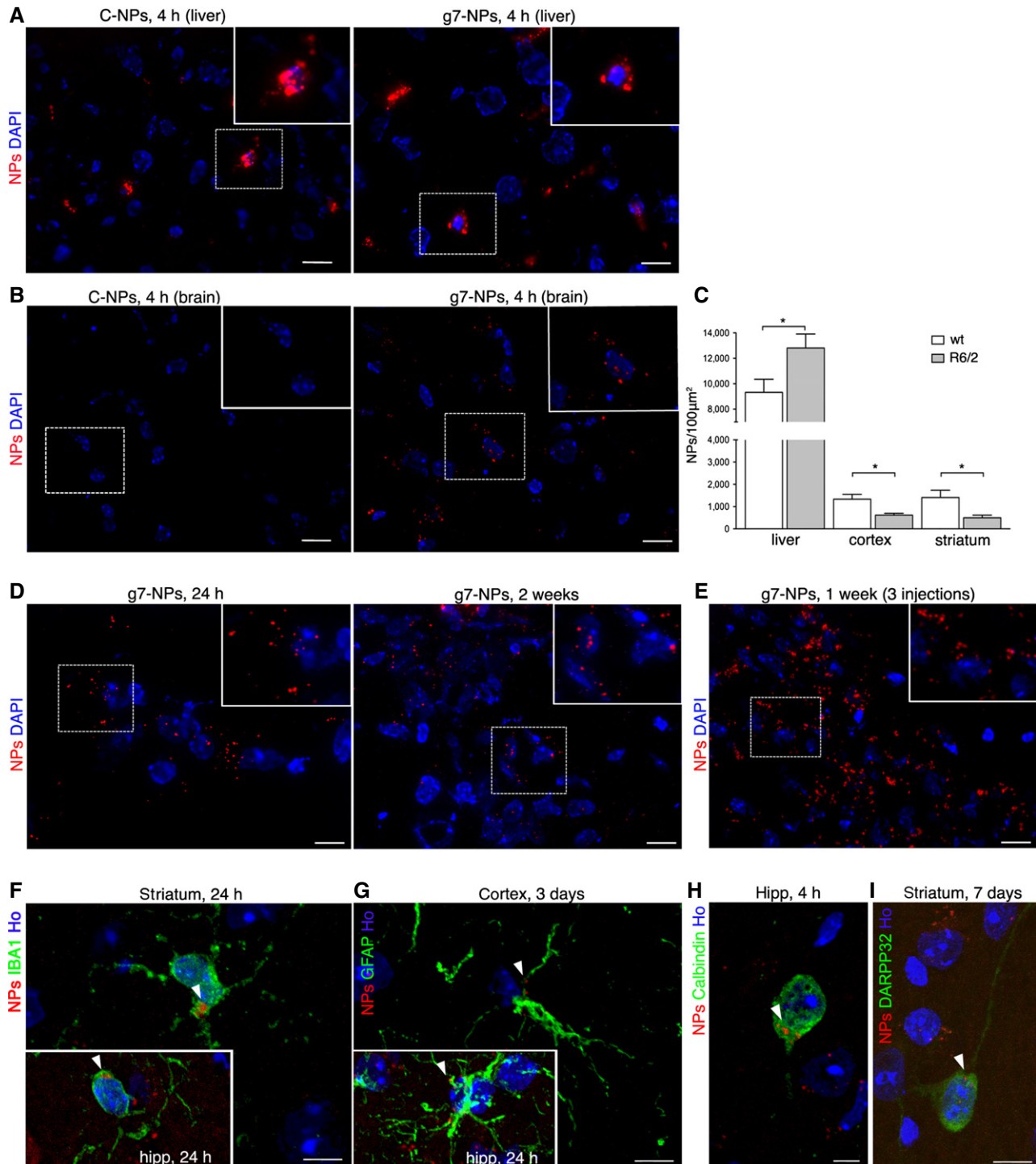

**Figure 2.  g7-NPs reach different brain cells and release cholesterol in R6/2 mice.**

A, B   Representative confocal images of liver (A) and brain (B) slices from R6/2 mice ip injected with C-NPs (left) or with g7-NPs (right) and sacrificed after 4 h.

C      Quantification of g7-NPs localized in the liver, striatum, and cortex of WT (*n* = 3) and R6/2 mice (*n* = 3). Data are expressed as the number of g7-NPs for 100 μm² ± SEM. Statistics: *\*P* < 0.05 determined by Student's *t*-test.

D, E   g7-NPs in brain slices from R6/2 mice administered with a single ip injection and sacrificed after 24 h (D, left) or 2 weeks (D, right) and after multiple ip injections within 1 week (E).

F–I    Representative confocal images of immunostaining against IBA1 (F), GFAP (G), calbindin (H), and DARPP-32 (I) on coronal sections of brains isolated from R6/2 mice ip injected with g7-NPs and sacrificed at the indicated time points. White arrowheads indicate intracellular g7-NPs.

Data information:  DAPI (A, B, D) or Hoechst 33258 (Ho) (F–I) was used to counterstain nuclei. Scale bars: 20 μm (A); 10 μm (B, D, E); 5 μm (F–I).

two experimental regimens. One group of R6/2 mice and WT littermates were administered 0.15 mg g7-NPs/g body weight for each injection, once every two weeks (three injections total) while the second experimental group was injected twice a week (ten injections total) accounting for a total estimated amount of 6.3 μg or 21 μg of cholesterol injected, respectively. Four groups were compared: WT mice administered with saline (referred to as WT), R6/2 mice administered with saline (referred to as R6/2), R6/2 injected with empty g7-NPs (referred to as R6/2-emp), and R6/2 mice receiving three or ten injections (referred to as R6/2-Chol). At sacrifice, the presence of g7-NPs was analyzed by fluorescence microscopy in liver sections from each animal and in cortical samples taken from the brains before the electrophysiological recordings (Appendix Fig S7). At the end of the analyses, data from the two experimental paradigms were pooled together as no significant differences were found.

HD mouse models have been extensively analyzed for their basic striatal electrophysiological phenotypes. Similar and robust defects have been described in striatal MSNs, namely a reduced membrane capacitance, a decrease in spontaneous excitatory postsynaptic current (EPSC), and an increase in spontaneous inhibitory post-synaptic current (IPSC) frequencies (Cepeda et al, 2003). The consistency of these phenotypes across different HD mouse models suggests that these changes are a result of the mutant huntingtin gene. We therefore tested whether our experimental scheme for cholesterol supplementation could reverse any of these phenotypes. Our whole-cell patch-clamp recordings of MSNs in brain slices showed that membrane capacitance, a reflection of membrane area, was significantly reduced in R6/2 mice treated with saline or empty g7-NPs (R6/2-untreated; data were pooled as no differences were found), compared to WT mice (WT, treated with saline) (Appendix Table S3). In contrast, in neurons from R6/2 mice treated with g7-NPs-Chol (R6/2-Chol), cell capacitance was not significantly reduced compared to WT cells, suggesting a mild rescue of cell membrane area (Appendix Table S3). Input resistance was found increased in both R6/2-untreated and R6/2-Chol neurons compared to WT neurons. Additionally, a significant decrease in the decay time constant in cells from R6/2-Chol mice compared with cells from WT or R6/2 mice treated with saline or empty g7-NPs was observed (Appendix Table S3). This effect may be attributed to changes in membrane fluidity induced by cholesterol supplementation.

The average frequency of spontaneous IPSCs was also significantly higher in MSNs from R6/2-untreated mice compared to WT mice (Fig 4A and B, inset), as previously observed (Cepeda et al, 2004). In contrast, R6/2-Chol mice displayed a significant reduction in the frequency of IPSCs compared to R6/2-untreated (Fig 4B, inset), in particular for small-amplitude events (< 40 pA; Fig 4B), while the cumulative inter-event interval histogram showed a decreased release probability in R6/2-Chol compared to R6/2-untreated cells (Fig 4C). Similar to MSNs from R6/2-untreated mice, IPSCs from R6/2-Chol mice had faster kinetics than cells from WT mice as judged by shorter decay time and half-amplitude duration of the current events compared to WTs (Appendix Table S4A).

The frequency of spontaneous excitatory postsynaptic currents (EPSCs) (Fig 4D) was significantly reduced in R6/2-untreated mice compared to WT mice (Fig 4E, inset). Although the decrease in

the average frequency of EPSCs was not significantly rescued in R6/2-Chol mice, the cumulative inter-event interval indicated a significantly increased release probability in R6/2-Chol cells versus R6/2-untreated cells (Fig 4F). EPSC kinetics was similar among groups, except for half-amplitude duration, which was significantly shorter in R6/2-chol cells than in WT cells (Appendix Table S4B). Altogether, these findings indicate that specific membrane and synaptic alterations observed in MSNs from R6/2 mice can be rescued by in vivo cholesterol supplementation through g7-NPs.

## Cholesterol supplementation ameliorates cognitive dysfunction in HD

We next assessed the impact of cholesterol supplementation on the behavior of HD mice by using motor and cognitive tasks. The injections regimen used in the behavioral studies, described in Fig 5A, is the same employed for electrophysiological studies. In the rotarod test, R6/2 mice treated with saline or empty g7-NPs exhibited typical impaired coordination compared to WT mice, as indicated by a shorter latency to fall from an accelerating rotarod. This deficit was not improved in R6/2-Chol mice (Fig 5B). Similarly, in the open field test, reduced rearing activity, which is a form of vertical exploration, was not rescued by cholesterol supplementation at 10 weeks of age (Fig 5C). At the same age, the hypokinetic phenotype shown in R6/2 mice (measured as global activity in the open field) was still apparent in R6/2-Chol, but the phenotype was less dramatic compared to R6/2-untreated mice and significance reached $P < 0.05$ (Fig 5D), suggesting that cholesterol supplementation partially ameliorates locomotion-related behavior in a novel environment. Other parameters (stereotyped movements, locomotion, resting time, mean velocity) showed similar changes (Appendix Fig S8A). Accordingly, R6/2-untreated mice worsened over time more than R6/2-Chol mice (Appendix Fig S8B) as indicated by the significant difference reached at later time points when the two groups were compared at 8 and 10 weeks of age. These findings suggest that sustained and repeated cholesterol supplementation might slow the disease progression.

As changes in cholesterol synthesis/levels are associated with cognitive decline (Suzuki et al, 2013), we next evaluated cognitive tasks in HD mice after cholesterol supplementation. To evaluate cognitive performance, we used the novel object recognition test, a low-stress task aimed at evaluating recognition memory. Importantly, object memory is impaired in patients with HD. In a pattern recognition task, subjects have to remember and touch the abstract patterns they are shown during training and that are paired with a novel pattern during testing. Early HD patients and clinically symptomatic subjects performed significantly worse than control subjects (Lawrence et al, 1996, 2000). R6/2-untreated mice showed a pronounced inability to discriminate novel from familiar objects from 8 weeks of age and worsened over time (Fig 5E). Notably, R6/2-Chol mice performed as well as WT mice, indicating that cholesterol supplementation rescued memory deficits at all time points (Fig 5E). Importantly, the time-course analysis also revealed that the benefit on recognition memory in R6/2-Chol mice was still present at 12 weeks of age, that is, 3 weeks after the last injection (Fig 5E).

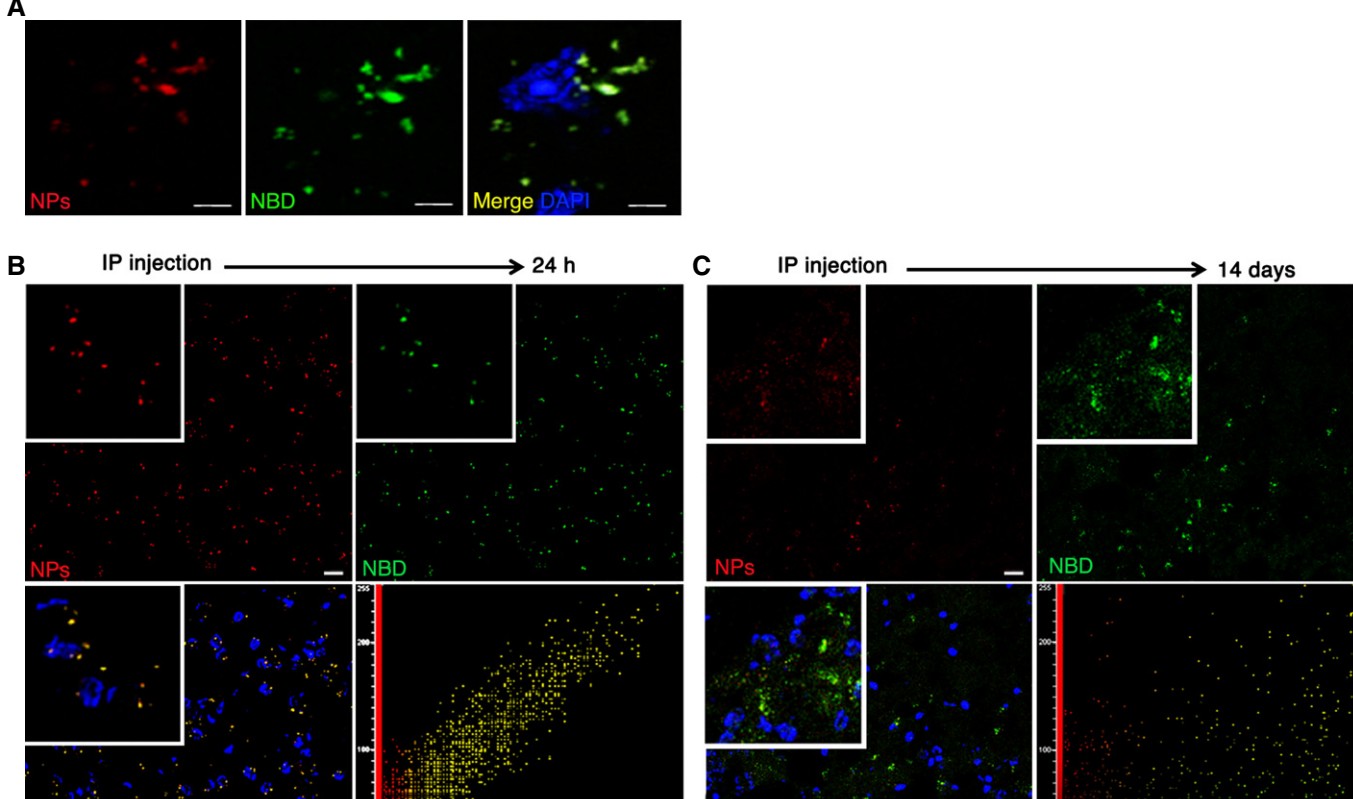

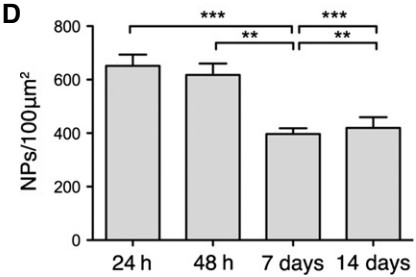

**Figure 3.  Cholesterol delivery and release *in vivo* in the R6/2 brain.**

A    Representative confocal image (crop) of brain slices from R6/2 mice ip injected with rhodamine-labeled g7-NPs-NBD-Chol and sacrificed after 12 h, showing co-localization of NBD-Chol (green) and rhodamine (NPs, red). Scale bar: 5 μm.

B, C    Representative confocal image (low magnification) of brain slices from R6/2 mice ip injected with g7-NPs-NBD-Chol and sacrificed after 24 h (B) or 2 weeks (C) and relative co-localization of NBD-Chol and g7-NPs. Scale bar: 10 μm.

D    g7-NPs quantification in brain slices at the same time points in (B, C). Data are expressed as number of g7-NPs (evaluated based their size) for 100 μm$^2$ ± SEM. Statistics: **$P < 0.01$ (48 h vs. 7 days; 7 days vs. 14 days), ***$P < 0.001$ (24 h vs. 7 days; 7 days vs. 14 days) determined by one-way ANOVA followed by Newman–Keuls multiple comparison test.

Data information: DAPI was used to counterstain nuclei.

## Cholesterol supplementation restores levels of synaptic components but not neuropathology

To determine whether cholesterol supplementation modulates synaptic protein machinery, we used biochemically purified triton-insoluble fractions (TIF) from the brain of WT ($n = 5$),

R6/2-untreated ($n = 6$) and R6/2-Chol ($n = 3$) mice and performed semiquantitative Western blotting for scaffolding proteins such as PSD95 and gephyrin and NMDA receptor subunits (GluN1 and GluN2B) (Fig 6A). Reduced PSD95, as well as a reduction in GluN1 and GluN2B, were found in R6/2-untreated mice compared to WTs, as expected (Fig 6B). Importantly, cholesterol supplementation

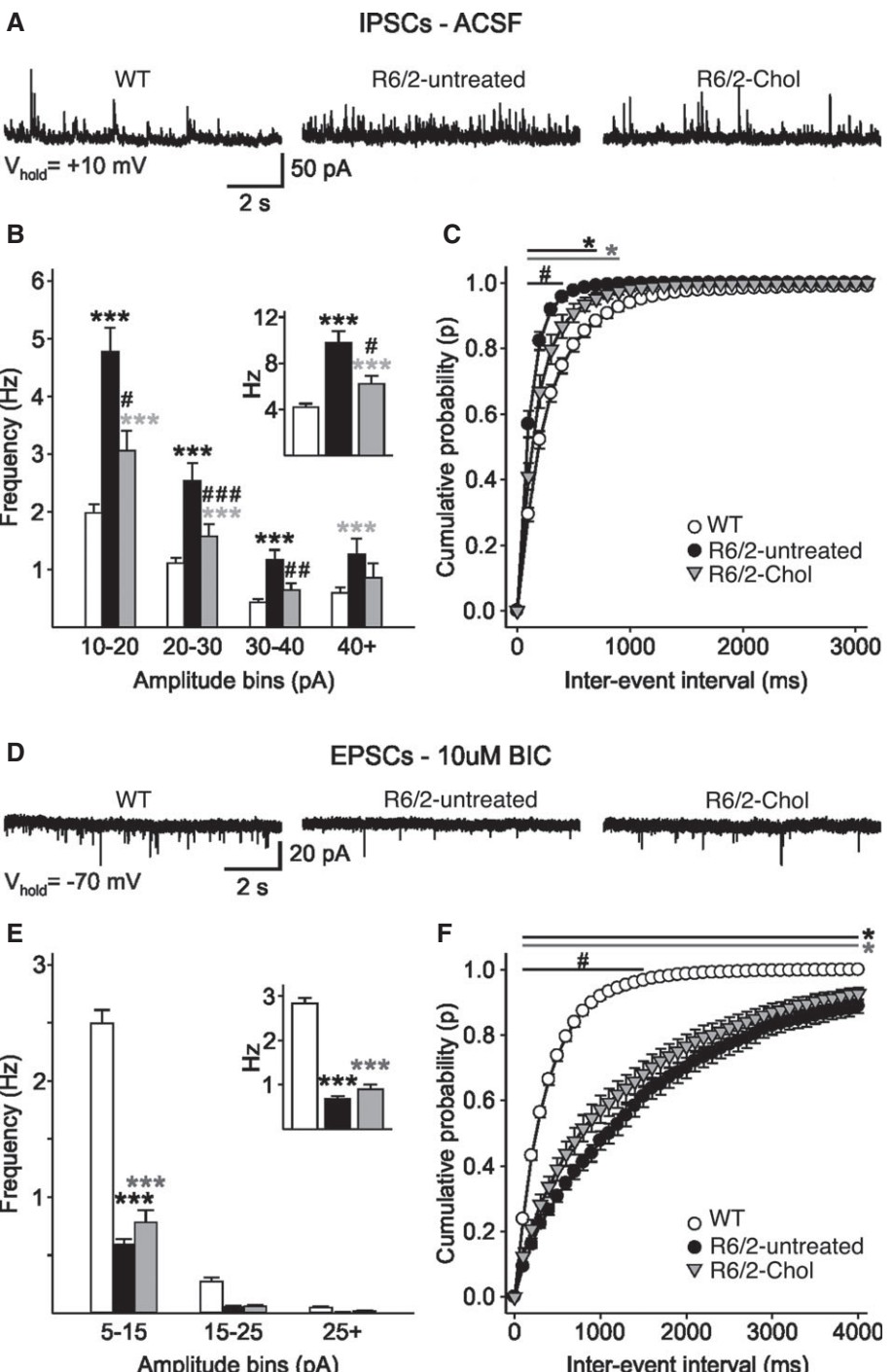

**Figure 4.  Systemic injections of g7-NPs-Chol rescue synaptic alteration in R6/2 mice.**

A   Spontaneous IPSCs were recorded from striatal MSNs (WTs = 52; R6/2-untreated = 27; R6/2-Chol = 29) at a holding potential of +10 mV. As no differences were found between R6/2 mice treated with saline (R6/2) or with empty g7-NPs (R6/2-emp), data were pooled.

B   Amplitude–frequency histogram and average frequency (inset) of IPSCs from R6/2-Chol, R6/2-untreated, and WT MSNs.

C   Cumulative inter-event histogram showing the release probability of IPSCs in all groups.

D   Spontaneous EPSCs were recorded from striatal MSNs (WTs = 52; R6/2-untreated = 27; R6/2-Chol = 29) at a holding potential of −70 mV. As no differences were found between R6/2 mice treated with saline (R6/2) and with empty g7-NPs (R6/2-emp), data were pooled.

E   Amplitude–frequency histogram and average frequency (inset) of EPSCs from R6/2-Chol, R6/2-untreated, and WT MSNs.

F   Cumulative inter-event histogram showing the release probability of EPSCs in all groups.

Data information: (B, D–F) Data represent mean ± SEM. $P < 0.05$ was determined by one-way ANOVA followed by Newman–Keuls multiple comparison tests ([#]$P < 0.05$, [##]$P < 0.01$, [###]$P < 0.001$ R6/2-untreated mice vs. R6/2-Chol mice; *$P < 0.05$, **$P < 0.01$, ***$P < 0.001$ WT mice vs R6/2-untreated or R6/2-Chol mice).

normalized or increased the levels of these proteins (Fig 6B), suggesting a rescue of the molecular composition of the synaptic machinery contributing to synaptic structure.

To evaluate whether cholesterol supplementation also influences the expression of synaptic genes, we performed qRT–PCR for a panel of synaptic genes known to be reduced in HD. BDNF mRNA is reduced in brain of several HD mice and is considered a critical hallmark in HD (Zuccato and Cattaneo (2014). Unpaired *t*-test between R6/2-untreated and R6/2-Chol groups revealed a slight but significant increase of *bdnf* expression in HD cortex after cholesterol supplementation (Fig 6C). We also evaluated the expression of *snap25* and *complexin II* (Fig 6D and E), the latter being a gene encoding for a presynaptic protein involved in neurotransmitter release (Reim *et al*, 2001). Similarly to *bdnf,* mRNA levels of snap25 were significantly increased in cortex of HD mice after cholesterol supplementation (Fig 6D). mRNA level of *complexin II* was strongly reduced in cortex, hippocampus, and striatum of R6/2-untreated mice compared to WTs. Cholesterol supplementation significantly increased *complexin II* expression in hippocampus and striatum from R6/2-Chol mice compared to R6/2-untreated mice (Fig 6E). These findings suggest that cholesterol supplementation partially ameliorates transcriptional abnormalities in the synaptic machinery in HD mice.

We also quantified mRNA levels of genes considered to be MSN markers, that is, darpp32, dopamine receptor D2 (drd2), and muscarinic acetylcholine receptor M4 (chmr4). As expected, all these genes were reduced in the striatum from R6/2-untreated mice compared to WTs, but cholesterol supplementation did not significantly influence their expression (Appendix Fig S9).

To investigate whether cholesterol supplementation counteracts striatal atrophy and MSN degeneration, we performed unbiased stereological analyses at 12 weeks of age. Reduced striatal volume and enlargement of ventricles, both measures of striatal atrophy, were observed in R6/2 mice treated with saline (R6/2) compared to WTs (Appendix Fig S10; Fig 6F), as already reported in the literature. The administration of empty g7-NPs or g7-NPs-Chol did not influence striatal volume in R6/2 mice (Appendix Fig S10). A statistically significant reduction in ventricular volume was evident in R6/2-Chol in comparison with R6/2 mice, similar to that observed in WTs (Fig 6F). However, R6/2 mice treated with empty g7-NPs (R6/2-emp) also showed a similar rescue, suggesting that the administration of g7-NPs *per se*, likely due to degradation of PLGA in lactic and glycolic acids, might influence this neuropathological parameter.

Altogether, these findings suggest that cholesterol supplementation via g7-NPs is not sufficient to counteract brain atrophy and neurodegeneration in R6/2 mice, at least with this experimental paradigm, although it does increase the expression of specific genes and synaptic proteins.

### *In vivo* evaluation of safety of g7-NPs in HD mice

Cholesterol supplementation to the brain might lead to a further reduction in cholesterol synthesis, already compromised in R6/2 mice (Valenza *et al*, 2007b). We therefore measured cholesterol precursors and the brain-specific cholesterol catabolite 24OHC in the brain of the treated mice at 12 weeks of age. Lathosterol, a marker of cholesterol synthesis, was equally reduced in both

R6/2-untreated and R6/2-Chol mice compared to WTs (Fig 7A), suggesting that exogenous cholesterol supplemented via g7-NPs does not further decrease the endogenous biosynthetic pathway. Similarly, 24OHC, an indicator of brain cholesterol catabolism that usually mirrors cholesterol biosynthesis in brain (Lund *et al*, 2003), was found similarly reduced in both R6/2 groups compared to WTs (Fig 7B).

As it is known that most of the NPs (90%) localize in periphery, we also measured mRNA levels of cholesterol biosynthetic genes (*hmgcr* and *fdft1*) in liver and lung. The mRNA expression of both cholesterol genes was similar in both tissues in all groups, even in the presence of g7-NPs-Chol (Fig 7C and D). All these results suggest that the exogenous cholesterol delivered to the brain or accumulated in peripheral tissues does not lead to alterations of endogenous cholesterol homeostasis in the time frame analyzed in this study.

Although the NPs employed in this study are considered biocompatible and biodegradable as made of PLGA, which is approved by the FDA and EMA, an immunogenicity study of these NPs *in vivo* is missing. Both PLGA, released after degradation of empty or cholesterol-loaded g7-NPs, and cholesterol itself might influence immune responses. Therefore, we analyzed mRNA levels of two pro-inflammation genes encoding for TNF-alpha and IL6, in peripheral tissues from our cohorts. As shown in Fig 7E and F, *Tnf-alpha* and *Il6* mRNA levels were significantly increased in the liver and in lung of R6/2 mice treated with saline (R6/2) compared to WTs, supporting the available evidence that peripheral inflammation is associated with HD condition (Trager *et al*, 2014; Chang *et al*, 2015). Similar activation of inflammatory genes was also observed in R6/2 mice treated with empty g7-NPs (R6/2-emp) or g7-NPs-Chol (R6/2-Chol), suggesting that multiple administrations of g7-NPs (empty or loaded with cholesterol) do not affect *per se* peripheral inflammation in R6/2 mice.

## Discussion

Synaptic dysfunction is an attractive target for possible HD therapies as it occurs early in the disease process when cell death in HD models is not obvious (Cepeda *et al*, 2004; Cummings *et al*, 2006; Joshi *et al*, 2009; Milnerwood *et al*, 2010) and paralleling the evidence that cognitive disturbances in patients with HD occur long before onset of overt motor manifestations (Levine *et al*, 2004; Paulsen *et al*, 2008; Schippling *et al*, 2009; Orth *et al*, 2010). We show that exogenous cholesterol supplementation to the HD mouse brain restores normal synaptic communication and protects mice from cognitive decline. This study provides the missing link between the reduction in brain cholesterol in the mouse HD brain and some of the neuronal abnormalities in the disease state. The data herein reported are in line with our recent *in vitro* studies, suggesting that strategies aimed at supplying cholesterol to HD neurons can ameliorate neuronal and synaptic dysfunction (Valenza *et al*, 2015).

Cholesterol supplementation via ip injection of cholesterol-loaded NPs normalizes GABAergic and, partially, glutamatergic synaptic activity in striatal MSNs of R6/2 mice (Fig 4), supporting the relevance of cholesterol in synaptic integrity and neuronal

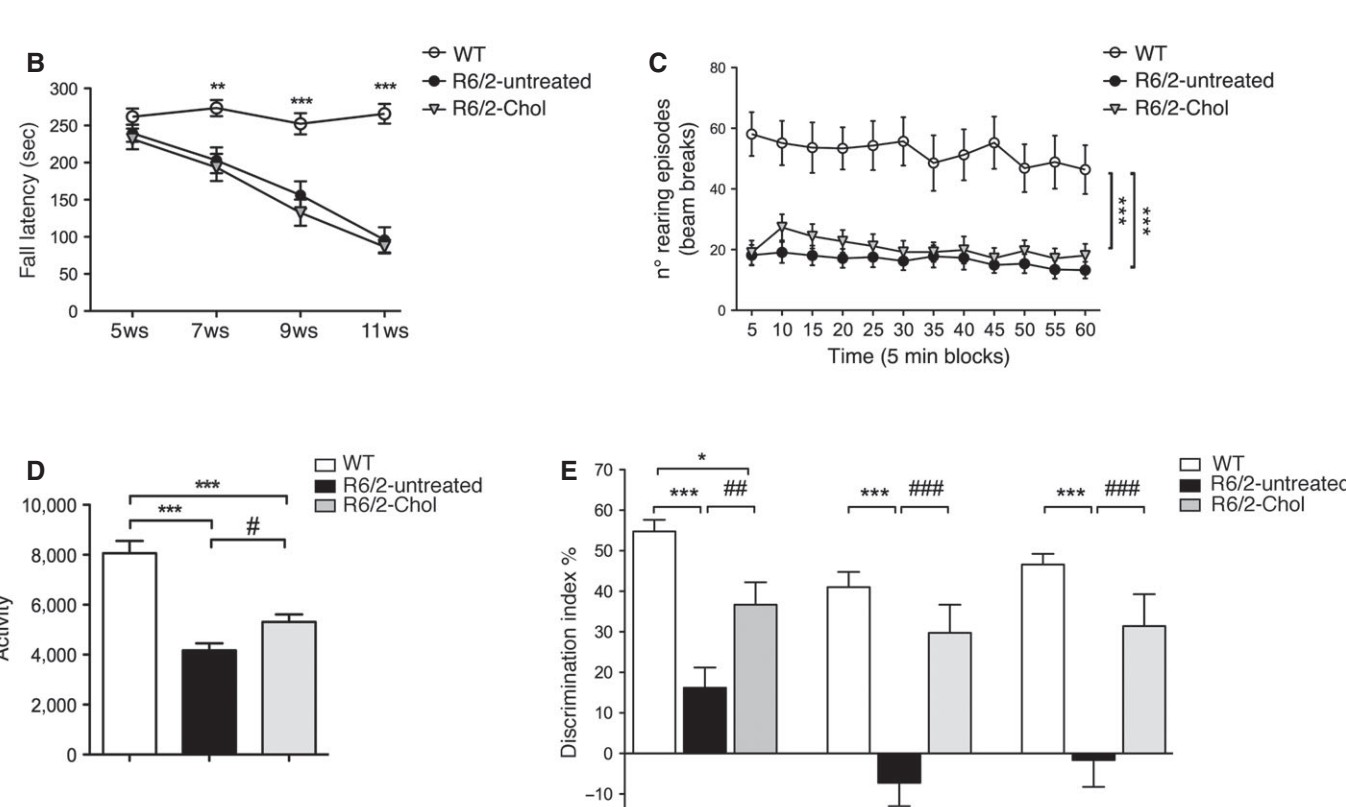

**Figure 5.   Systemic injections of g7-NPs-Chol ameliorate cognitive defects in R6/2 mice.**

A   Experimental paradigm performed in all the trials.
B   Fall latency from an accelerating rotarod for 5- to 11-week-old WT and R6/2 mice during cholesterol supplementation (WT = 17; R6/2-untreated = 21; R6/2-Chol = 13).
C   Rearing activity in open field at 10 weeks of age (WT = 14; R6/2-untreated = 21; R6/2-Chol = 15).
D   Global activity in the open field test at 10 weeks of age (WT = 14; R6/2-untreated = 21; R6/2-Chol = 15).
E   Index of discrimination (%) in WT, R6/2-untreated, and R6/2-Chol mice during disease progression, at 8 weeks of age (WT = 24; R6/2-untreated = 36; R6/2-Chol = 21), at 10 weeks of age (WT = 25; R6/2-untreated = 35; R6/2-Chol = 20), and at 12 weeks of age (WT = 24; R6/2-untreated = 30; R6/2-Chol = 19); the index above zero indicates a preference for the novel object; the index below zero indicates a preference for the familiar object. As no differences were found between R6/2 mice treated with saline (R6/2) or treated with empty g7-NPs (R6/2-emp), data were pooled.

Data information: Data in (B–E) are from three independent trials and represent mean ± SEM. $P < 0.05$ was determined by two-way ANOVA (in B, C) and by one-way ANOVA (in D, E) followed by Newmann–Keuls multiple comparison tests ([#]$P < 0.05$, [##]$P < 0.01$, [###]$P < 0.001$ R6/2-untreated mice vs. R6/2-Chol mice; [*]$P < 0.05$, [**]$P < 0.01$, [***]$P < 0.001$ WT mice vs. R6/2-untreated or R6/2-Chol mice).

function (Pfrieger, 2003). Cholesterol supplementation also protects R6/2 mice from cognitive decline as measured by recognition memory recovery (Fig 5E). Novel object preference has been often associated with the hippocampus; however, it also depends on functional interaction between hippocampus and cortex (Barker & Warburton, 2011) and, more recently, it has been linked to the striatum (Darvas & Palmiter, 2009). Consistently, reduction of the cholesterol sensor SCAP in the brains of mice leads to a decrease in brain cholesterol synthesis and causes impaired synaptic transmission and altered cognitive function assessed by novel object

recognition test (Suzuki *et al*, 2013). Of note, cognitive decline has been recently associated with hippocampal cholesterol loss and cholesterol infusion in aged mice improved learning and memory in aged rodents (Martin *et al*, 2014a).

Several *in vitro* findings indicate that synaptic transmission is sensitive to cholesterol levels both at pre-synaptic and post-synaptic levels. Indeed, cholesterol depletion affects vesicle recycling and fusion (Thiele *et al*, 2000; Dason *et al*, 2010, 2014; Linetti *et al*, 2010), AMPARs mobility (Hering *et al*, 2003; Renner *et al*, 2009; Martin *et al*, 2014b), and the distribution and

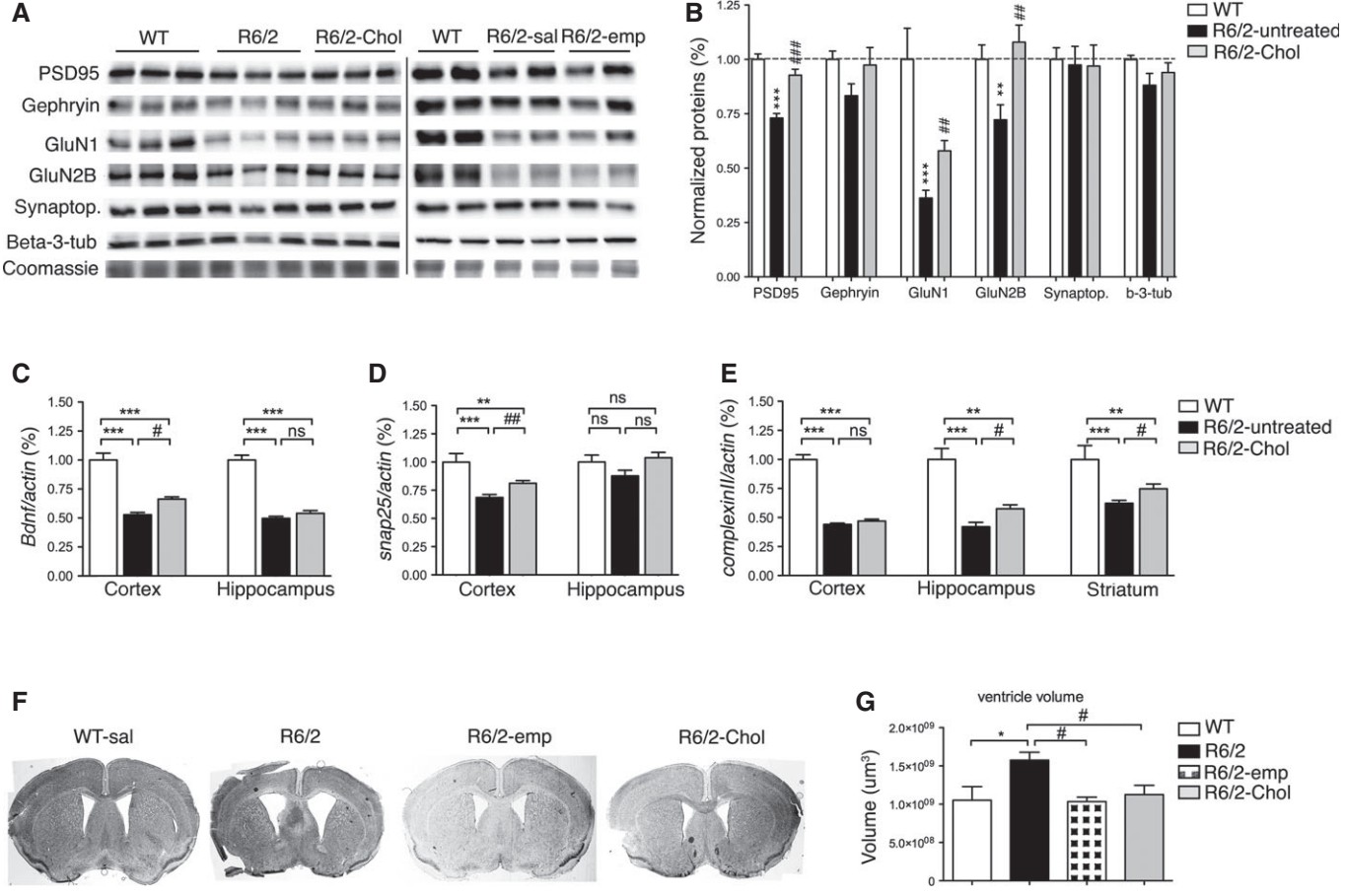

**Figure 6.  Systemic injections of g7-NPs-Chol positively influence synaptic protein network but not neuropathology.**

A, B   Protein levels (A) and relative densitometry quantification (B) of several synaptic proteins in triton-insoluble (synaptic enriched) fractions purified from total brains from WT (*n* = 5), R6/2-untreated (*n* = 5) and R6/2-Chol (*n* = 3). Levels of PSD95 and NMDA receptor subunits GluN1 and GluN2B are rescued in R6/2 mice by cholesterol supplementation.

C–E   mRNA levels for Bdnf (C), and Snap25 (D) in cortex and hippocampus; Complexin II (E) in cortex, hippocampus, and striatum from a subset of WT (*n* = 4), R6/2-untreated (*n* = 7), and R6/2-chol animals (*n* = 3). As no differences were found between R6/2 mice treated with saline or treated with empty g7-NPs, data were pooled.

F, G   Representative images of Nissl staining (F) and ventricle volume revealed by Neurolucida analysis at 12 weeks of age in WT (*n* = 7), R6/2 (*n* = 7), R6/2-emp (*n* = 6), and R6/2-Chol (*n* = 8) mice.

Data information: Data in (B–E, G) represent mean ± SEM. $P < 0.05$ was determined by one-way ANOVA followed by Newman–Keuls multiple comparison tests (in B, E) and by Student's *t*-test between R6/2-untreated and R6/2-Chol (in C, D) (#*P* < 0.05, ##*P* < 0.01, ###*P* < 0.001 R6/2-untreated mice vs. R6/2-Chol mice; *\**P* < 0.05, *\*\*P* < 0.01, *\*\*\*P* < 0.001 WT mice vs. R6/2-untreated or R6/2-Chol mice).

Source data are available online for this figure.

---

function of NMDAR (Frank *et al*, 2004, 2008). Accordingly, we found that cholesterol supplementation increases the levels of the scaffold synaptic protein PSD95 and NMDARs in synaptic protein-enriched fractions of HD mice (Fig 6A and B), suggesting that *in vivo* delivery of cholesterol contributes to preserve the structure and integrity of the synaptic machinery. In agreement with the biochemical findings, the partial but significant increase of mRNA levels of *bdnf, snap25,* and *complexin II* (all involved in synaptic transmission) in different brain regions of HD mice after cholesterol supplementation (Fig 6C–E) suggests that cholesterol may act at different levels in improving synaptic and cognitive functions. In particular, mRNA levels of complexin II, a key player in the mechanisms underlying cognitive processes (Reim

*et al*, 2001; Glynn *et al*, 2003), are reduced in R6/2 mice and in human HD striatum and cortex (Morton & Edwardson, 2001; Freeman & Morton, 2004) and complexin II knockout mice show selective cognitive deficits that reflect those seen in R6/2 mice (Glynn *et al*, 2003). The cognitive benefits might be also related to an effect of exogenous cholesterol on hormone steroids (Hara *et al*, 2015), but further studies are needed to address this issue.

Cholesterol is also required to establish proper membrane permeability, fluidity, and thickness, and it stabilizes membranes and provides order to membranes. The partial rescue of membrane capacitance and the decrease in the decay time constant that we have observed in striatal MSNs from R6/2 mice treated with g7-NPs-Chol (Appendix Table S4) suggest that cholesterol supplementation

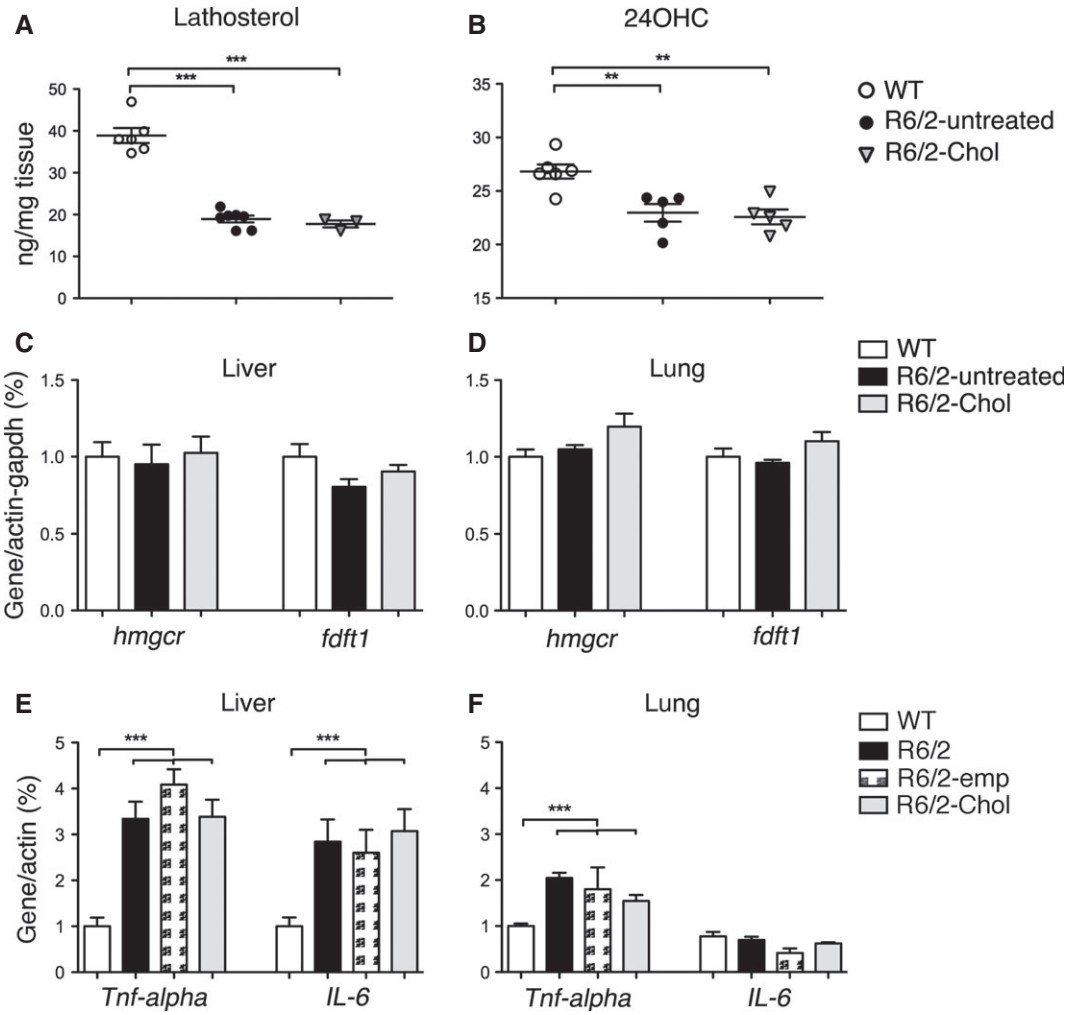

**Figure 7. Exogenous cholesterol or g7-NPs *per se* do not influence endogenous cholesterol synthesis or induce inflammatory response.**

A, B   Lathosterol and 24OHC measured by mass spectrometry in the brain of WT (*n* = 6), R6/2-untreated (*n* = 7) and R6/2-Chol (*n* = 5) mice.
C, D   mRNA levels of hmgcr and fdft1 in liver and lung of WT (*n* = 7), R6/2-untreated (*n* = 8) and R6/2-Chol (*n* = 4) mice. As no differences were found between R6/2 mice treated with saline or treated with empty g7-NPs, data were pooled.
E, F   mRNA levels of inflammatory genes in liver and lung of the of WT (*n* = 7), R6/2 (*n* = 4), R6/2-emp (*n* = 4) and R6/2-Chol (*n* = 5) mice.

Data information: Data in (A–F) represent mean ± SEM. *P < 0.05, **P < 0.01, ***P < 0.001 was determined by one-way ANOVA followed by Newman–Keuls multiple comparison tests.

induces changes in membrane fluidity. However, other than showing its incorporation into the membrane (Appendix Fig S5), we were not able to establish where exactly the exogenous cholesterol, once released by g7-NPs, localizes in brain cells, and more refined methods to visualize exogenous cholesterol at subcellular levels are needed.

We also reported that cholesterol supplementation does not rescue motor defects and restores only partially the global activity of HD mice (Fig 5B and D). The dichotomy that we observed in rescuing cognitive but not motor functions might be associated with specific roles of cholesterol in neuronal function. Similarly, neuropathological hallmarks such as striatal volume and MSN markers (Appendix Figs S9 and S10) do not significantly change after cholesterol supplementation in R6/2 mice, suggesting that

cholesterol alone, at least within this experimental paradigm, is not sufficient to prevent brain atrophy or to improve neuropathology. Of note, both empty g7-NPs and g7-NPs-Chol seem to counteract the enlargement of ventricle volume observed in R6/2 mice (Fig 6F and G) likely due to degradation of PLGA in lactic and glycolic acids that might influence metabolic pathways related to energy. However, the significance of this effect is unknown and cannot be ascribed to cholesterol. We should also consider that the content of cholesterol delivered to the brain through g7-NPs is estimated to be 21 μg (for the ten injections) and, although we started the treatment at 5 weeks of age, cholesterol was released after 1–2 weeks as shown by the co-localization studies (Fig 3). Both low dose and slow timing of cholesterol release might not be sufficient to reverse motor performance and brain atrophy in HD mice. Further studies, by

employing mini-pump-based strategies for early and continuous delivery of well-defined concentrations of cholesterol, will allow to gain more information about the possible impact of cholesterol delivery on motor defects.

Recent advances in nanotechnology and growing needs in biochemical applications have driven the development of multifunctional nanoparticles. Nanodrugs in liposome- or albumin-based formulations are already used in the clinic for some forms of cancers (Barenholz, 2012; Sethi *et al*, 2013; Von Hoff *et al*, 2013), and others are being tested in pre-clinical trials (Tasciotti *et al*, 2008; Shen *et al*, 2013). This is the first study in which g7-NPs have been applied to a disease model for CNS targeting of molecules that are not able to cross the BBB (such as cholesterol). g7-NPs reach different brain regions 2–4 h after a single systemic injection and localize in different brain cells, including striatal neurons (Figs 1 and 2). Previous pharmacological and biodistribution studies estimated that the percentage of g7-NPs that reaches the brain is > 10% of the injected dose (Tosi *et al*, 2007) and that multiple non-receptor-mediated mechanisms are implicated (Tosi *et al*, 2011b). Other NPs carrying ligands, antibodies, or peptides for specific receptors that enter into the brain by receptor-mediated endocytosis usually reach the brain compartment as maximum level values ranging from 0.1 to 1% of the injected dose (Gabathuler, 2010; Tosi *et al*, 2012; Gosk *et al*, 2004) owing to a possible saturation of the receptor or by the competiveness of endogenous ligands. Therefore, g7-NPs may represent a novel tool that can be used for brain delivery of several molecules. However, from a therapeutic prospective, we are conscious that additional quantitative studies of g7-NPs alone and loaded with cholesterol are needed to increase the knowledge about biodistribution and pharmacokinetics.

The polymeric NPs used in this study are made of PLGA, a copolymer approved by the FDA as drug delivery system for parenteral administration (Danhier *et al*, 2012). PLGA is considered biodegradable and biocompatible as it degrades completely into its original monomers, lactic and glycolic acid, which are easily metabolized in the body via the Krebs cycle and then eliminated (Shive & Anderson, 1997). However, specific studies concerning immune reactivity of g7-NPs are missing. Similarly, depending on the molecule delivered, specific studies should be performed in order to exclude any immune reaction or other side effects in different tissues. Our studies suggest that g7-NPs and cholesterol itself do not induce inflammatory response in liver and lungs, where almost all g7-NPs are localized. A more extensive biochemical study to evaluate the impact of g7-NPs degradation (and of the molecule released) is needed to accelerate preclinical testing and translational developments of these NPs.

A limitation of g7-NPs in this current study is the low drug loading (1%) that does not allow the delivery of elevated amounts of cholesterol. Presumably, a high amount of cholesterol, its strong affinity for hydrophobic interactions, and the rigidity of the sterane ring lead to the disruption of the PLGA organization as observed for NPs-Chol3 formation (Fig 1), with a marked increase in sample heterogeneity and low quality of nanoparticles. The identification of strategies aimed at increasing the amount of cholesterol encapsulated into g7-NPs without affecting chemical–physical properties of NPs will allow to reduce the number of injections/week while increasing the amount of cholesterol that reaches the brain cells. From another

prospective, the low content and the slow release of cholesterol by g7-NPs might be advantageous as cholesterol accumulation is dangerous for the brain. Further studies are needed to identify the threshold of cholesterol increase that is beneficial for HD brain/neurons and beyond which negative effects may occur.

The very low dose of exogenous cholesterol delivered in the brain of HD mice in our experimental paradigm (21 µg) does not allow to discriminate it from the large content of endogenous cholesterol even by mass spectrometry. However, the demonstration of a rescue in specific electrophysiological and behavioral phenotypes support the notion that the exogenous cholesterol delivered to the adult brain is sufficient to ameliorate neuronal dysfunction in HD. A similar concentration of cholesterol infused via osmotic pumps in aged mice has recently been able to improve learning and memory in aged rodents (Martin *et al*, 2014a).

In conclusion, these results emphasize the beneficial effects of cholesterol supplementation in reversing synaptic alterations and delaying cognitive defects in the HD mouse brain. Additionally, this study demonstrates the validity of a new technology based on g7-NPs to administer drugs (besides cholesterol) to the HD brain and lays the ground for future therapeutic approaches.

## Materials and Methods

### NPs formulation and characterization

Gly-L-Phe-D-Thr-Gly-L-Phe-L-Leu-L-Ser(O-β-D-Glucose)-CONH$_2$ (g7) was prepared as previously described (Tosi *et al*, 2011b) and conjugated with PLGA to obtain g7-PLGA. PLGA derivatization yields were confirmed by nuclear magnetic resonance to be 30–40 µmol peptide/g of polymer. PLGA conjugated with rhodamine (Sigma-Aldrich) was prepared as previously described (Costantino *et al*, 2005; Tosi *et al*, 2005). In all NPs, a fraction of polyvinyl alcohol (PVA) (about 12.5 mg PVA/100 mg NPs) remains stably associated with the NPs despite the repeated purification. The residual PVA forms a connected network with the PLGA chains becoming a "secondary" constituent of the NPs and partially masking the exposed acidic groups of the polymer. This explanation justifies the less negative values of ζ-pot with respect to those of the NPs prepared in the absence of PVA. Details related to the production and characterization of NPs and related to Fig 1 are listed in the Appendix Supplementary Methods.

### Cell culture and glial and neuronal differentiation

Neural stem (NS) cells carrying normal (Q7/7) or mutant htt (Q140/7) employed in this study and protocols for their differentiation were previously described (Conforti *et al*, 2013; Valenza *et al*, 2015). Primary neuronal cultures were prepared from the cortex of R6/2 mice embryos (day 18 of gestation) as previously described (Valenza *et al*, 2015).

### Animals and treatments

Experiments at the University of Milan were carried out in accordance with the European Communities Council Directive

2010/63/EU revising Directive 86/609/EEC regarding the care and use of animals for experimental procedures. All procedures at UCLA were performed in accordance with the U.S. Public Health Service Guide for Care and Use of Laboratory Animals and were approved by the Institutional Animal Care and Use Committee at UCLA. Genotyping of R6/2 mouse colonies (~150 CAG repeats) was performed by PCR of DNA obtained from tail samples, once at weaning and again following sacrifice for verification. The lifespan of this R6/2 mouse colony is approximately 12–14 weeks, with HD-like phenotypes evident from 8 weeks of age. All the mice have been randomly assigned to experimental groups, and the investigators have been blinded to the sample group allocation during the treatments and experiments. For each injection, the mice were administered 0.15 mg g7-NPs/g body weight (NPs stock concentration is 12.5 mg/ml; 0.7 mg in 100 mg of NPs), which corresponds to 1 µg of cholesterol/g. The chemical–physical characterization and drug content in the g7-NPs used in the pre-clinical trials is summarized in Appendix Table S5. The complete list of WT and R6/2 animals used for each experiment is described in Appendix Table S6. An initial trial was performed in WT and R6/2 mice treated with control NPs loaded with cholesterol (without g7, i.e., not able to cross the BBB). No changes were found in terms of behavioral tasks and molecular signature (Appendix Fig S11). Therefore, we decided to not include these groups in subsequent trials with g7-NPs-Chol.

### Immunohistochemistry

The animals were deeply anesthetized and transcardially perfused with 4% PFA. When only NPs were detected, cells or tissues were fixed in cold methanol at −20°C for 10 min, since fixation with paraformaldehyde reduced rhodamine-related NP fluorescence. Immunohistochemistry was performed on 15–30 µm coronal sections with the following primary antibodies: rabbit anti-IBA1 (1:500; Wako), rabbit anti-GFAP (1:250; Dako), rabbit anti-calbindin28 kDa (1:100; Swant), mouse anti-DARPP32 or rabbit anti-DARPP32 (1:100, Epitomics; S. Cruz), and mouse anti-PMCA ATPase (clone 5F10, 1:500; Thermo Scientific). Alexa Fluor 488-conjugated goat secondary antibodies (1:1,000; Invitrogen) were used for detection. Sections were counterstained with the nuclear dye Hoechst 33258 or 4′,6-diamidino-2-phenylindole (DAPI) (Invitrogen). Confocal images were acquired with a ZEISS LSM 510 or a LEICA SP5 laser scanning confocal microscopes.

### NPs quantification

To quantify NPs in different tissues, we used ImageJ software to measure the fluorescence derived from the rhodamine used to label the NPs. NPs were counted in 10 images for each tissue taken from three WT and three R6/2 mice. Images were divided into three color channels to set a threshold for the red produced by the NPs, and we calculated the percentage of red signal for each image. Knowing the total area of the field and the size of the NPs, we calculated the approximate number of NPs in the selected area. The count of NPs in the liver and in the brain was made at 20× and 60×, respectively, and the data were normalized to compare the results. Ten images for each animal/condition were analyzed. The images were acquired with a Leica AF6000LX microscope.

### Electrophysiology

At 10–11 weeks of age, mice were anesthetized with isoflurane and decapitated, and the brain was rapidly removed to ice-cold dissection artificial cerebrospinal fluid (ACSF) containing 130 mM NaCl, 3 mM KCl, 26 mM NaHCO$_3$, 1.25 mM NaHPO$_4$, 10 mM glucose, 5 mM MgCl$_2$, and 1 mM CaCl$_2$ oxygenated with 95% O$_2$/5% CO$_2$ (pH 7.2–7.4, osmolality 290–310 mOsm/l). Coronal slices (300 µm) of the striatum were cut with a microtome (Model VT 1000S, Leica Microsystems) and transferred to an incubating chamber containing oxygenated standard ACSF (with 2 mM CaCl$_2$ and 2 mM MgCl$_2$) for 1 h before electrophysiological recordings.

Whole-cell patch-clamp recordings were obtained from MSNs visualized in slices with the aid of infrared video microscopy and identified by somatic size and basic membrane properties (membrane capacitance, input resistance, and time constant). The patch pipette (3–5 MΩ) was filled with solution containing 125 mM Cs-methanesulfonate, 4 mM NaCl, 3 mM KCl, 1 mM MgCl$_2$, 9 mM EGTA, 8 mM HEPES, 5 mM MgATP, 1 mM Tris-GTP, 10 mM disodium phosphocreatine, and 0.1 mM leupeptin (pH 7.2, osmolality 270–280 mOsm/l).

Spontaneous postsynaptic currents were recorded in standard ACSF. The membrane current was filtered at 1 kHz and digitized at 100–200 µs using Clampex 10.2 (gap-free mode). Cells were voltage-clamped at −70 mV to assess basic membrane properties. Membranes were stepped to a holding potential of +10 mV to assess GABA$_A$ receptor-mediated IPSCs. Bicuculline methiodide (10 µM) was added to block GABA$_A$ receptor-mediated currents, and spontaneous glutamate receptor-mediated EPSCs were recorded at a holding potential of −70 mV. Spontaneous synaptic currents and event kinetics were analyzed offline using the automatic detection protocol within the MiniAnalysis Program (Synaptosoft) and checked manually for accuracy. Event counts were performed blind to genotype and treatment. The threshold amplitude for the detection of an event (5 pA for glutamatergic currents and 10 pA for GABAergic currents) was set above the root mean square background noise level (1–2 pA at V$_{hold}$ = −70 mV and 2–3 pA at V$_{hold}$ = +10 mV). Amplitude–frequency and inter-event interval distributions were constructed to evaluate differences in events at each amplitude and interval.

### Behavioral characterization

Rotarod: Mice were first trained at a fixed speed of 4 rpm on the apparatus (model 47600, Ugo Basile). After 1 h, the mice were tested in an accelerating task (from 4 to 40 rpm) over 5 min, for three trials per day for three consecutive days with an inter-trial interval of 30 min. Latency to fall was recorded for each trial and averaged. Open Field: The animals were placed individually into the center of a transparent, square, activity-cage arena (45 cm × 45 cm) (2Biological Instrument). Both horizontal and vertical activities were assessed, monitoring mice allowed to freely move for 60 min using the Actitrack software (2Biological Instrument) connected to infrared sensors placed all around the square cage. Novel Object Recognition Test: The device consisted of a Plexiglass square arena (dimensions: 40 × 40 × 40 cm). All phases of the test were conducted in the presence of low-intensity light. Mice were first habituated to the arena in the absence of objects for 15 min (on one

day, in the morning). On the same day, in the afternoon, two similar objects were presented to each mouse for 10 min (A′ and A″), after which the mice were returned to their home cage. Twenty-four hours later, the same animals were tested for 10 min in the arena with a familiar object (A″) and a new object (B). The index of discrimination was calculated as (time exploring the novel object − time exploring the familiar object) / (time exploring both objects) × 100. Object preference was measured as (time exploring each object) / (time exploring both objects) × 100. All experiments were done blind to genotypes.

### Triton-insoluble protein fraction preparation and Western blot

Triton-insoluble fractions of the brain were prepared as described in the study by Gardoni *et al* (2009), separated on SDS–PAGE and probed with specific antibodies. Antibodies used in these experiments include anti-PSD-95 (1:1,000; #124011 SySy), NMDAR1 (GluN1) (1:500; #AB9864, Millipore), NMDAR2B (GluN2B) (1:500; #MAB57578, Millipore), gephyrin (1:1,000; #147111, SySy), synaptophysin (1:1,000, S. Cruz), and beta-3-tubulin (1:3,000; #G7121, Promega). Horseradish peroxidase-conjugated secondary antibodies were then used (1:3,000; Bio-Rad). Bands were visualized with enhanced chemoluminescence (Pierce) and imaged with the ChemiDoc MP Imaging System (Bio-Rad). The bands were densitometrically quantified (Image Lab, Bio-Rad) and normalized for Coomassie staining. Beta-3-tubulin was used as an additional loading control.

### RNA isolation, retrotranscription, and real-time quantitative PCR

Total RNA from tissues was isolated with TRIzol reagent (Life Technologies). Total RNA (0.25–1 μg) was reverse-transcribed to single-stranded cDNA using the iScript cDNA synthesis kit (Bio-Rad). For each reverse-transcribed product, three real-time PCR analyses were performed in duplicate for each of the analyzed genes. An iCycler thermal cycler with a Multicolor Real-time PCR Detection System (Bio-Rad) was used to evaluate gene expressions. Taqman probes with a FAM dye label (for cholesterol genes) or EVA Green Supermix (for inflammatory genes) was used, as previously described (Valenza *et al*, 2015).

### Nissl staining and Neurolucida analysis

Animals were perfused and brains dissected, frozen and serially cut (30 μm-thick coronal sections) on the cryostat. One 30-μm-thick section every five was stained with cresyl violet (Nissl staining). Briefly, sections were dried overnight. Then, they were dehydrated with a scale alcoholic of ethanol and xylene, than rehydrated, and immersed in 1% cresyl violet and 1% glacial acetic acid aqueous solution for 5 min. The staining was followed by a new dehydration in ascending ethanol and xylene. Sections were cover-slipped with Leica CV mounting media (Cat#14046430011). Brain, ventricle, and striatum perimeters (relative to one hemisphere) were reconstructed in a cerebral segment included between plates 19 and 39 of the Franklin K. and Paxinos G. atlas (Paxinos & Franklin, 2008). They were drawn at 40× at a microscope with a motorized stage interfaced to the computer, using the Neurolucida software (Microbrightfield Inc., VT, USA). The obtained volumes were analyzed with the

---

**The paper explained**

**Problem**

Huntington's disease is a genetic neurodegenerative disorder characterized by progressive motor, cognitive, and psychiatric disturbances. Cholesterol biosynthesis and content are reduced in the brain of multiple animal models of HD. This dysfunction—of cerebral origin—is measurable in blood of patients with HD since pre-symptomatic stages of disease. However, a link between reduced synthesis/level of cholesterol and neuronal dysfunction *in vivo* in HD is missing. As circulating or dietary cholesterol is not able to cross the blood–brain barrier (BBB) and cholesterol in the brain depends largely on endogenous biosynthesis, this dysfunction may be detrimental for neuronal function especially given that locally synthesized cholesterol is implicated in synapses formation, integrity, and remodeling.

**Results**

To address the relationship between cholesterol dysfunction and synaptic and cognitive deficits in HD mouse models, we delivered cholesterol into the brain by using a novel technology based on cholesterol-loaded polymeric nanoparticles further modified with a peptide (g7) to cross the BBB after systemic injection in the mice. We showed that these nanoparticles (g7-NPs) reach different brain regions and different brain cells and gradually release cholesterol after their degradation. We also showed that repeated systemic administration of cholesterol-loaded g7-NPs in HD mice: (i) rescues synaptic communication in striatal medium-sized spiny neurons, (ii) prevents cognitive decline and partially improves global activity, and (iii) restores the levels of proteins that compose the synaptic machinery.

**Impact**

Neuronal and synaptic dysfunction is an attractive target for possible HD therapies because it occurs long before cell death in mouse models and in humans with HD. An intervention at this stage could, in theory, slow or stop neuron loss before it starts. Our conclusions highlight the relevance of cholesterol deficits in cognitive impairment associated with HD and the benefits of cholesterol supplementation with a broad impact for other brain disorders.

In parallel, the evidence that g7-NPs can be used as vectors for the delivery of therapeutic molecules (besides cholesterol) to the brain opens new and medically very relevant scenarios for the treatment of several CNS disorders. Importantly, the nanoparticles employed are made of PLGA, which is approved by FDA in various drug delivery systems in humans as it is considered biodegradable and biocompatible.

---

Neuroexplorer software (Microbrightfield Inc.) using the Cavalieri formula for volume reconstruction.

### Measurement of sterols

Samples were prepared and analyzed by isotopic dilution mass spectrometry as previously described (Valenza *et al*, 2010).

### Statistics

SigmaPlot 12.3 (Systat software) or Prism 5 (GraphPad software) was used to perform all statistical analyses. Data are presented as means ± standard error of the mean (SEM). Grubbs' test was applied to identify outliers. For each set of data to be compared, we determined in Prism whether data were normally distributed or not. As they were all normally distributed, we used parametric tests.

Indeed, differences between group means were assessed with an unpaired Student's *t*-test, and two-way or one-way ANOVA followed by Bonferroni or Newman–Keuls *post hoc* tests, as indicated in the text. Differences were considered statistically significant if $P < 0.05$. No statistical methods were used to pre-determine sample sizes, but our sample sizes are similar to those reported in the literature. For details, see also Appendix Table S7 showing statistical analyses and *P*-values for the main figures.

**Expanded View** for this article is available online.

## Acknowledgements

We thank Luca Pignata and Chiara Orciani for technical assistance and Elisa Battaglia for help with the behavioral tests. We also thank Miriam Ascagni (CIMA, an advanced microscopy facility established by Università degli Studi Milano, Milan), Valeria Berno and Silvia Tartari (Imaging Facility in INGM, Milan), and Centro Grandi Strumenti (University of Modena and Reggio Emilia) for technical assistance and support with confocal analysis. This work was partially supported by Neuromics European grant (305121) to E.C. and by Ministero della Salute under 40 (GR-2008-1145270) to M.V and V.L.. The electrophysiology studies were supported by USPHS NS41574, HD004612, and NS081335 to M.S.L. and C.C. The Hereditary Disease Foundation (to M.L. and G.T.) supported the preparation of g7-NPs for the electrophysiological studies.

This paper is dedicated to the memory of our dear colleague, devoted scientist and well known neurologist Stefano Di Donato.

## Author contributions

EC and MV developed the study, conceived the experimental plans, and analyzed the data. JYC, CCe, and MSL developed and performed the electrophysiological experiments and analyzed the data. GT, BR, DB, MAV, and FF developed the NP-based strategy, produced, and characterized all NPs used in this work. MV, EB, and GT performed the immunostaining experiments and provided most of the confocal images. MV performed biochemical analyses and with CFB all the molecular analyses; MV, EDP, and CFB performed the behavioral tests. VL, CCa and SDD performed all the mass spectrometry analyses. MMB and AV performed all reconstruction analyses with Neurolucida. MV, GT, BR, and EC interpreted the data and wrote the manuscript. All authors read and edited the manuscript. EC supervised the entire work and gave final approval of the manuscript version to be published.

## Conflict of interest

The authors declare that they have no conflict of interest.

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
