## [Review Process File · EMBO Molecular Medicine]

Cholesterol-loaded nanoparticles ameliorate synaptic and cognitive function in Huntington's disease mice

Marta Valenza, Jane Y. Chen, Eleonora Di Paolo, Barbara Ruozi, Daniela Belletti, Costanza Ferrari Bardile, Valerio Leoni, Claudio Caccia, Elisa Brillì, Stefano Di Donato, Marina M. Boido, Alessandro Vercelli, Maria A. Vandelli, Flavio Forni, Carlos Cepeda, Michael S. Levine, Giovanni Tosi, Elena Cattaneo

Corresponding author: Elena Cattaneo, University of Milan

Review timeline:

Submission date:	06 May 2015
Editorial Decision:	16 June 2015
Revision received:	16 September 2015
Editorial Decision:	07 October 2015
Revision received:	20 October 2015

Transaction Report:

Editor: Céline Carret

1st Editorial Decision

16 June 2015

Thank you for the submission of your manuscript to EMBO Molecular Medicine and please accept my apologies for the delay in getting back to you. All three referees had grant deadlines and I was attending a meeting which further delayed the decision making. I have now heard back from these three referees and while they find the study to be of interest, they also raise a number of concerns that need to be addressed in the next version of your article.

You will see from the comments below that referees 2 and 3 are supportive of publication, even though referee 2 asks for additional controls and validation that would certainly increase the conclusiveness of the findings. Referee 1 is more reserved, especially regarding the choice of the mouse model. After consulting with the other two referees about this issue, while they both agree that confirming the data in a different model would certainly greatly improve the paper, they however do not find this request absolutely required for evaluation of the revised manuscript. Therefore, we would be happy to consider a revised manuscript providing that all other issues are properly addressed, and experimentally when suggested.

Please note that it is EMBO Molecular Medicine policy to allow only a single round of revision and that, as acceptance or rejection of the manuscript will depend on another round of review, your responses should be as complete as possible.

Revised manuscripts should be submitted within three months of a request for revision; they will otherwise be treated as new submissions, except under exceptional circumstances in which a short extension is obtained from the editor. Please follow the recommendations below for submission of

your revised article.

I look forward to seeing a revised form of your manuscript as soon as possible.

***** Reviewer's comments *****

Referee #1 (Comments on Novelty/Model System):

see remarks below

Referee #1 (Remarks):

The EMM-2015-05413 manuscript by Valenza and colleagues aims at testing the effect of cholesterol supplementation as a therapeutic approach for Huntington's disease (HD). The authors use nanoparticles modified with glycopeptides (g7NPs) and loaded with cholesterol to treat R6/2 mice. The authors confirmed that g7NPs release cholesterol *in vivo* in the brain of R6/2 animals. They next determine the impact of cholesterol supplementation on different parameters by performing electrophysiology experiments, behavioral tests and biochemical and gene expression evaluations of synaptic components and other neuronal markers.

This work is a follow-up of several publications on the same subject matter by this group. Overall the topic of the study is highly significant. Indeed, HD is a disorder for which no treatment is available at the current time. Brain cholesterol homeostasis is altered in HD. The understanding of the underlying mechanisms of such dysregulation and targeting dysregulated pathways as therapeutic opportunities is of great interest for the field. The use of nanoparticles to deliver treatment is also most certainly important.

This being said, I have major concerns with the study: not on the experimental level (the experiments are in general well conducted) but on the general design (see specific comments below). Furthermore, the supplementation of cholesterol had little or no effect on the different parameters tested except for the amelioration of the cognitive defects. I thus feel that this manuscript lacks the depth required for publication in EMBO Molecular Medicine.

Major concerns with the design of the study :

1) Why did the authors choose R6/2 as a model of HD? R6/2 mice have been widely used in the HD field providing critical information. It is still appropriate for specific questions. However, expression of short N-terminal fragments of mutant huntingtin does not take into account several aspects of HD, in particular those related to the biology of the protein (an aspect that has been one of the focus of this lab). Over the last 15 years, none of the many screens performed on R6/2 mice led to a drug inducing a therapeutic benefit in patients. Mice expressing full-length mutant huntingtin would have been more relevant to model HD for this study.

2) The rationale of using supplementation of cholesterol as a therapeutic approach in HD is based on the observation that cholesterol biosynthesis and levels are reduced in HD mouse models and this is indeed the case in the R6/2 model. However, different groups have reported conflicting results in various HD models. The authors completely ignore this literature in the current version of their manuscript. Notably, accumulation of free cholesterol at the plasma membrane was observed in HD situation. In fact, the two observations (decrease in some models and increase at the plasma membrane in others) could be explained by the fact that most cholesterol synthesis occurs in glia, thus the reduced cholesterol synthesis observed may reflect cholesterol level in these cells. In neurons, the situation may be different. Furthermore, the subcellular distribution of cholesterol has to be considered (a decrease in intracellular cholesterol could be coupled to an increase of lipid rafts cholesterol). This should be at least discussed by the authors (and the work of others should be cited). But it also raises doubts on the use of cholesterol supplementation as a treatment for HD.

3) This doubt is reinforced by the observation that the treatment has little or almost no effect on the different parameters evaluated. The authors can not state that they rescue synaptic alteration given the amplitude of the effect. The only marked benefit of cholesterol treatment is on cognitive impairment.

4) I am not sure how to interpret the fact that R6/2 mice treated with empty g7-NPs show a rescue (here there is a complete rescue) of the ventricle volume see Figure 6F. This makes it difficult to interpret the other observations too.

Referee #2 (Comments on Novelty/Model System):

Very good paper, though needs to address two critical issues: i) define whether beneficial effects of peripherally injected cholesterol in the HD mouse are due to a central (CNS) or peripheral (peripheral organ hormone production) mechanism and ii) prove of brain cells' membrane incorporation of exogenous cholesterol.

Referee #2 (Remarks):

This is a most interesting piece of work, with a strong (if not exclusive) translational value: biodegradable nanoparticles (NPs) linked to the g7 peptide, for more efficient brain penetration, loaded with cholesterol result in the reduction of a number of pathological signs in the R6/2 Huntington mouse model. The work is of high technical value. Authors have performed electron and atomic force microscopy for the characterization of the size and shape of the nanoparticles, quantification of levels of NP in the brain and other organs in wild-type and R6/2 mouse, localization to different types of brain cells, levels of association to, and diffusion of, cholesterol and phenotypic effects and toxicity studies. Before acceptance authors need to address two critical points:

i) define whether the beneficial effects are truly produced by the restoration of cholesterol in brain cells or to a brain non-autonomous effect due to increased steroid hormone production by peripheral organs concentrating NP-cholesterol (gonads, adrenal glands). It is in fact well known that steroid hormones can modulate cognition and synaptic plasticity.

ii) provide a better (quantitative) demonstration that peripheral exogenous cholesterol is truly incorporated on brain cells' membranes. Am afraid the NBD signal is not sufficient proof of membrane incorporation.

Injection of cholesterol nanoparticles in the cisterna magna or one of the brain ventricles might be a good experimental approach to address both questions: a). it would yield sufficient levels of NP-cholesterol to quantify exogenous membrane cholesterol incorporation and b) rule out peripheral organ contribution (unlikely that brain injected NP-cholesterol reaches peripheral organs in levels high enough to result in increased steroid hormone production). A simpler experiment would be to measure the different steroid hormones levels in the mice after systemic injection of the g7-chol NPs. Ideally, authors should include a 5th (wt) and 6th (R6/2) groups of mice for the rescue (electrophysiology and behavior) studies: cholesterol nanoparticles without the g7 carrier (no brain entry of cholesterol).

I appreciate the value of the IP treatments as a promising strategy in HD yet find somewhat disturbing not knowing if the effect is brain autonomous or not and if the peripheral cholesterol becomes inserted on brain cells' membranes.

Referee #3 (Remarks):

This manuscript is built on previous interesting results from the authors showing deficits in cholesterol metabolism in the neurodegenerative Huntington disease (HD). Here they have developed glucopeptidommodified nanoparticles to deliver cholesterol into a commonly used mouse model for the disease, the R6/2 mouse. They convincingly show that these particles cross the BBB and that the product is being taken up by both glial and neuronal cells. They show that this treatment has a positive effect on synaptic function, general activity as well as cognitive function in the mice.

This is a very well and carefully performed study that is written in a clear and logical fashion. The

results are well presented, and the interpretation and discussion of the results are well balanced. The data from this study holds promise for new treatment possibilities for HD although much research is still needed in order to further develop this therapy and improve its efficacy.

1st Revision - authors' response

16 September 2015

Referee #1 (Remarks):

The EMM-2015-05413 manuscript by Valenza and colleagues aims at testing the effect of cholesterol supplementation as a therapeutic approach for Huntington's disease (HD). The authors use nanoparticles modified with glycopeptides (g7NPs) and loaded with cholesterol to treat R6/2 mice. The authors confirmed that g7NPs release cholesterol in vivo in the brain of R6/2 animals. They next determine the impact of cholesterol supplementation on different parameters by performing electrophysiology experiments, behavioral tests and biochemical and gene expression evaluations of synaptic components and other neuronal markers.

This work is a follow-up of several publications on the same subject matter by this group. Overall the topic of the study is highly significant. Indeed, HD is a disorder for which no treatment is available at the current time. Brain cholesterol homeostasis is altered in HD. The understanding of the underlying mechanisms of such dysregulation and targeting dysregulated pathways as therapeutic opportunities is of great interest for the field. The use of nanoparticles to deliver treatment is also most certainly important.

We thank the referee for his/her comments.

This being said, I have major concerns with the study: not on the experimental level (the experiments are in general well conducted) but on the general design (see specific comments below). Furthermore, the supplementation of cholesterol had little or no effect on the different parameters tested except for the amelioration of the cognitive defects. I thus feel that this manuscript lacks the depth required for publication in EMBO Molecular Medicine.

Major concerns with the design of the study:

1) Why did the authors choose R6/2 as a model of HD? R6/2 mice have been widely used in the HD field providing critical informations. It is still appropriate for specific questions. However, expression of short N-terminal fragments of mutant huntingtin does not take in account several aspects of HD, in particular those related to the biology of the protein (an aspect that has been one of the focus of this lab). Over the last 15 years, none of the many screens performed on R6/2 mice led to a drug inducing a therapeutic benefit in patients. Mice expressing full-length mutant huntingtin would have been more relevant to model HD for this study.

We understand well the point raised by the reviewer but we are not aware of compounds tested in other HD models that have led to a drug. Respectfully, we believe that it is too early to exclude animal models and, on the contrary, we need them all. There are in fact problems also with mice expressing full-length mutant huntingtin. To our view, the failure is often related to the general difficulties in translating any animal study into human and not to a given animal model. Hence our approach has always been to test a given finding in multiple animal models. Furthermore, the US National Institute of Neurological Disorders and Stroke developed recommendations for improving preclinical trials pointing to rigorous study designs and on how to report the results of animal research in manuscripts (Landis et al., Nature 2012). These recommendations are critical for solid pre-clinical trials and we are following these guidelines. Hence, the study presented in the submitted paper is our first attempt in delivering cholesterol loaded NPs but we do want to extend the study to more animal models, an effort that we hope to accomplish in the next several years.

Having said that, we are also perfectly aware of the criticisms in the use of R6/2 mouse model, which are mostly due to the early manifestation and fast progression of the phenotype. Our reading of the literature, however, highlights that there is no perfect model that recapitulates every aspect of the human disease (Levine et al., 2004; Pouladi et al., 2014). In that sense, the same criticism can be raised for other models. For example, in the most commonly used full-length models (e.g., YAC128, BACHD) the progression of the phenotype is very slow or atrophy and cell loss quite absent. In addition, full-length HD mice gain weight, a finding that is not typically seen in the human condition.

Accordingly, the reasons for us to choose the R6/2 model are the following:

- a) It was the first model created and the most widely used. To date, over 500 articles have been published using this model, more than with any other model, including full-length or knock-in models.
- b) It has become the gold standard for drug testing (Gil and Rego, 2008), creating the basis for future studies in additional models.
- c) When background strain and disease progression are taken into account, R6/2 and knock-in mice (150 CAG repeats) show consistent changes in molecular phenotypes (Woodman et al., 2007).
- d) Due to its fast progression, it is more amenable to electrophysiological studies as, for technical reasons, visualization of individual neurons and quality of the recordings are always better in brain tissue from younger animals. In addition, studies have shown that synaptic abnormalities are consistent among R6/2, full-length and knock-in models (Cummings et al., 2010).
- e) Multiple rodent models show the same alteration in cholesterol biosynthesis found in R6/2. So far, cholesterol dysfunction has been validated in at least six HD animal models (R6/2, YAC128, YAC72, YAC46, Q111 mice and HD rats; Valenza et al., 2005; 2007a, 2007b, 2010) and, more recently, we have measured *de-novo* synthesis of cholesterol in heterozygous knock-in Q175 mouse brain, in parallel to our classical isotopic dilution mass spectrometry measurements (unpublished; manuscript in preparation) with the same results as in R6/2 mice. In all animals so far tested, cholesterol biosynthesis is reduced in the brain before the onset of motor defects and cholesterol content is significantly decreased at later time points, a finding that we have first reported in the R6/2 mouse line.

2) The rationale of using supplementation of cholesterol as a therapeutic approach in HD is based on the observation that cholesterol biosynthesis and levels are reduced in HD mouse models and this is indeed the case in the R6/2 model. However, different groups have reported conflicting results in various HD models. The authors completely ignore this literature in the current version of their manuscript. Notably, accumulation of free cholesterol at the plasma membrane was observed in HD situation.

We are aware that a controversy has emerged in the past on this subject but we hope that it is now solved by taking into account a number of considerations as detailed in Marullo et al., 2012. Detection of cholesterol by colorimetric (i.e. filipin staining) and enzymatic assays (based on cholesterol esterase and oxidase) should be supplemented with more sensitive analytical methods (mass spectrometry) and care must be taken to prepare the sample appropriately. Due to space constraints we did not include a discussion on the subject as it was extensively discussed in our previous papers and reviews. In short, accumulation of free cholesterol was reported in HD mouse brains and in HD cells by Trushina et al., 2006, Del Toro et al., 2010 and Luthi-Carter et al., 2010. However, in all those cases, colorimetric and enzymatic methods, with low sensitivity, were used to detect cholesterol and the analyses were performed in protein and not in lipid extracts.

On the contrary:

- a) By adopting analytical approaches based on isotopic dilution mass spectrometry, in combination with other biochemical and molecular analyses, we have shown that cholesterol synthesis (in terms of lathosterol levels) in the brain is reduced not only in the R6/2 model but also in many other HD

models leading to a decrease of cholesterol content (see Valenza et al., 2007a, 2007b, 2010). Of note, in 2014, a decrease of lathosterol and cholesterol levels was found in the striatum of a knock-in HD mouse model (Q150) as judged by mass spectrometry (Trushina et al., 2014; table 1) by the same authors who previously found cholesterol accumulation with filipin staining (Trushina et al., 2006).

b) More recently, we have performed a cross-validation study in which samples from different brain regions of the heterozygous knock-in mouse model carrying 175 CAG repeats (Q175) at different disease stages were processed independently by two research units (us in Milan and another group in US) to quantify (i) the *de novo* cholesterol synthesis rate by $^2\text{H}_2\text{O}$ labeling *in vivo* and (ii) the concentrations of lathosterol (the main cholesterol precursor and indicator of cholesterol biosynthesis), cholesterol and catabolite 24-hydroxy-cholesterol (24OHC; the brain-specific cholesterol) by isotopic dilution mass spectroscopy. We found that the daily synthesis rate of cholesterol and the corresponding concentration of lathosterol are significantly reduced in the striatum of Q175 mice since the pre-symptomatic stage of disease (manuscript in preparation).

c) Other authors have reported that brain cholesterol homeostasis may be affected also in HD patients as concentration of plasma 24OHC, the brain-specific catabolite of cholesterol, is reduced early in HD patients and in pre-symptomatic individuals who are close to the onset of the disease. Of note, reduced plasma 24OHC correlates to reduced caudate volume observed in HD patients (Leoni et al., 2008; Leoni et al., 2013).

d) Although we cannot review again all these findings and the source of the controversial data we have now added one sentence in the introduction (pag. 3 in the revised text) to clarify the state of the art on the subject/controversy.

In fact, the two observations (decrease in some models and increase at the plasma membrane in others) could be explained by the fact that most cholesterol synthesis occurs in glia, thus the reduced cholesterol synthesis observed may reflect cholesterol level in these cells. In neurons, the situation may be different.

Furthermore, the subcellular distribution of cholesterol has to be considered (a decrease in intracellular cholesterol could be coupled to an increase of lipid rafts cholesterol). This should be at least discussed by the authors (and the work of others should be cited). But it also raises doubts on the use of cholesterol supplementation as a treatment for HD.

We agree with the reviewer. There is plenty of evidence showing that most of the cholesterol synthesis occurs in glia. Recently we have demonstrated that defective cholesterol synthesis in astrocytes is detrimental for HD neurons (Valenza et al., 2015). There is also evidence showing that changes in subcellular cholesterol level may occur in HD neurons (for example, HTT is associated with lipid rafts and this association is stronger for mutant than for wt HTT and might influence plasma cholesterol; Valencia et al., 2010).

It is possible that HD neurons have reduced/altered capability to handle changes in cholesterol levels that leads to an accumulation of intracellular cholesterol that cannot be used for neuronal functions. However, this hypothesis has not been demonstrated *in vivo* and we are not aware of data showing accumulation of cholesterol in neuronal membranes except for the *in vitro* data showing an increased filipin staining in the plasma membrane and cytosol in primary HD neurons with respect to wt neurons (Trushina et al., 2006; Del Toro et al., 2010). Filipin is an antibiotic polyene widely used as a histochemical marker for cholesterol, however, the specificity of cholesterol binding remains highly speculative (Gimpl and Gehrig-Burger 2007). We are aware that filipin staining cannot be considered a quantitative method for measuring sterol/cholesterol (as described in Marullo et al., 2012).

Of note, both reduced production and supply of cholesterol from astrocytes to neurons (as demonstrated in Valenza et al., 2015 by using quantitative methods) and a potential accumulation of free cholesterol in cytoplasm and membranes of neurons (as suggested in Trushina et al., 2006,2013;

Luthi-Carter et al., 2010; Del Toro et al., 2010 by using filipin staining) indicate that there is reduced availability of cholesterol for neuronal function, underlining – in both cases - the rationale for cholesterol supplementation as a treatment for HD.

3) This doubt is reinforced by the observation that the treatment has little or almost no effect on the different parameters evaluated. The authors can not state that they rescue synaptic alteration given the amplitude of the effect. The only marked benefice of cholesterol treatment is on cognitive impairment.

HD is a complex disease and is characterized by motor, cognitive and behavioral abnormalities. In this work we found that cholesterol supplementation via g7-NPs prevents cognitive defects (i.e. NORT), reduces the pathological increase in GABA synaptic activity and increases the levels of key synaptic proteins in synaptic-enriched subcellular fractions. The effect of cholesterol supplementation on these parameters was robust and highly significant. All these findings go in the same direction, indicating a beneficial effect on parameters related to synaptic activity and function.

We did not find a significant improvement in motor skills or in neuropathological signs (i.e. striatal volume or MSN markers). This might be due to the low concentration of cholesterol delivered to the brain via g7-NPs or to the cognitive aspects of the disease being preferential targets of a cholesterol supplementation therapy. To address this issue, we are performing pre-clinical trials by infusing cholesterol directly into the striatum of HD mice with mini-pumps and results will be presented in a separate paper.

In general, to our view, if it sounds surprising that cholesterol may affect only cognitive parameters, than it may be even more surprising (and unlikely) that a single molecule will be sufficient to rescue all HD phenotypes (motor and cognitive defects, synaptic activity abnormalities, neuropathological signs etc). In our case, we found a specific rescue of the cognitive aspects of the disease and this is in agreement with papers indicating a role of cholesterol in synaptic transmission, synaptogenesis, recycling of synaptic vesicles and in cognition (Allen et al., 2007, Martin et al., 2014). Finally, progressive cognitive impairment is the cause of major disability in HD (reviewed in Ross et al., 2014). Therefore, the therapeutical value of rescuing “only” the cognitive side of the disease should not be underestimated.

4) I am not sure how to interpret the fact that R6/2 mice treated with empty g7-NPs show a rescue (here there is a complete rescue) of the ventricle volume see Figure 6F. This makes it difficult to interpret the other observations too.

We agree with the referee that it is difficult to interpret the rescue of ventricle volume also with empty g7-NPs. One explanation may be related to the fact that the nanoparticles used in this work are made of poly(lactic-co-glycolic acid) (PLGA). Once NPs are degraded, the breakdown products, lactic acid and glycolic acid, are metabolized in the Krebs cycle (Anderson and Shive 1997). Of note, lactate, a product of lactic acid, is an efficient energy substrate and is preferentially metabolized by neurons in several mammalian species (including mice, rats and humans). Therefore, NPs are biocompatible but they are not inert after their degradation and may influence metabolic pathways related to energy. We have added a sentence in the Discussion Section to explain this issue (pag. 11 in the revised text).

Previous and more recent works identified metabolic defects in HD patients and HD models, suggesting a lack of substrates for the Krebs cycle (Mochel et al., 2007; Gouarnè and Pruss 2013). In addition, it has been reported that supplementation of energy substrates (i.e. creatine) may have neuroprotective effects and lead to an increase in striatal lactate concentration in vivo in HD models (Matthews et al., 1998). Although the role of creatine in HD patients is still debated after results from different clinical trials (<http://huntingtonstudygroup.org/tag/crest-e/>; Rosas et al., 2014; Herch et al., 2006; Tabrizi et al., 2003), g7-NPs *per se* might supply additional substrates for energy metabolism and have some effect.

Referee #2 (Comments on Novelty/Model System):

Very good paper, though needs to address two critical issues: i) define whether beneficial effects of peripherally injected cholesterol in the HD mouse are due to a central (CNS) or peripheral (peripheral organ hormone production) mechanism and ii) prove of brain cells' membrane incorporation of exogenous cholesterol.

Referee #2 (Remarks):

This is a most interesting piece of work, with a strong (if not exclusive) translational value: biodegradable nanoparticles (NPs) linked to the g7 peptide, for more efficient brain penetration, loaded with cholesterol result in the reduction of a number of pathological signs in the R6/2 Huntington mouse model. The work is of high technical value. Authors have performed electron and atomic force microscopy for the characterization of the size and shape of the nanoparticles, quantification of levels of NP in the brain and other organs in wild-type and R6/2 mouse, localization to different types of brain cells, levels of association to, and diffusion of, cholesterol and phenotypic effects and toxicity studies. Before acceptance authors need to address two critical points:

i) define whether the beneficial effects are truly produced by the restoration of cholesterol in brain cells or to a brain non-autonomous effect due to increased steroid hormone production by peripheral organs concentrating NP-cholesterol (gonads, adrenal glands). It is in fact well known that steroid hormones can modulate cognition and synaptic plasticity.

We evaluated the kinetics and the distribution of g7-NPs in different brain regions and in a subset of peripheral tissues but not in gonads and in adrenal glands. In all trials performed (n=12, five of which for behavior analyses; see Supplementary Table S6), we did not isolate these tissues, thus a potential study to evaluate the distribution and kinetics of g7-NPs or steroids hormones levels would require additional trials. As for sure these experiments will have to be repeated in more animal models we will be able to address this interesting point in future studies.

ii) provide a better (quantitative) demonstration that peripheral exogenous cholesterol is truly incorporated on brain cells' membranes. Am afraid the NBD signal is not sufficient proof of membrane incorporation.

Injection of cholesterol nanoparticles in the cisterna magna or one of the brain ventricles might be a good experimental approach to address both questions: a) it would yield sufficient levels of NP-cholesterol to quantify exogenous membrane cholesterol incorporation and b) rule out peripheral organ contribution (unlikely that brain injected NP-cholesterol reaches peripheral organs in levels high enough to result in increased steroid hormone production).

As suggested by the referee, we injected fluorescent cholesterol (25-NBD-cholesterol) into the brain ventricles and we performed a co-localization study between cholesterol and markers of plasma membrane (PMCA-ATPase) to test the ability of NBD-Chol to localize into the brain cells' membrane. We decided to use NBD-cholesterol because it closely resembles the structure of native cholesterol and allows us to visualize it by means of fluorescence. Although the referee says that NBD signal is not sufficient proof of membrane incorporation, 25-NBD-cholesterol is a valid and well-used tool to study cholesterol trafficking because it is incorporated into membranes and effectively probes cholesterol-containing domains compared to other fluorescent cholesterols (Gimpl et al., 2007; Daniel et al., 2010). Accordingly, our new results show that NBD-Chol co-localizes with PMCA-ATPase suggesting its incorporation into the brain cells' membrane (**new Supplementary Figure S5**, described at pag. 6 in the revised text).

We had also planned to use g7-NPs loaded with cholesterol labeled with deuterium to perform a

better quantification of cholesterol delivered by g7-NPs. Radioactive cholesterol should be more sensitive but we do not have the permissions to use radioactive both for the preparation of NPs and to treat the animals with radioactive molecules. We are now setting the condition to quantify by mass spectrometry very low concentrations of deuterated cholesterol and to discriminate it from the endogenous cholesterol in tissues. However, the preparation of g7-NPs loaded with deuterated cholesterol (including their characterization and their chemical-physical properties), the treatment of mice and tissue analysis, will require many months. We hope that the reviewer will agree with us on the fact that this specific study should be part of a separate manuscript, in which we will further verify the nanoparticles-based strategy by including rigorous pharmacokinetic/dynamic studies in different tissues and subcellular fractions, therefore creating the basis for subsequent more translational studies.

A simpler experiment would be to measure the different steroid hormones levels in the mice after systemic injection of the g7-chol NPs.

We appreciate this comment. We can't exclude an effect of cholesterol on steroids and we have added a sentence about this issue in the Discussion section (pag. 10 in the revised text). It is likely that cholesterol acts at multiple levels either directly or as a substrate for downstream pathways. The link between cholesterol biosynthesis (and cholesterol supplementation in HD brain) and steroids is however far from being established and future studies are needed to address this issue. Lipidomics/metabolomics studies could help evaluating the impact of cholesterol dysfunction (and cholesterol supplementation) on different metabolic pathways in central and peripheral tissues. However, we believe this is beyond the scope of this manuscript.

As a further comment, we believe that the effects we observe are more consistent with central effects. Neuroactive steroids are potent modulators of GABA_A receptor activity and are potent sedatives and anesthetics. A sedative effect would be unlikely to ameliorate cognitive deficits. Also, neurosteroids are positive allosteric modulators of GABA_A receptors, which potentiate the actions of GABA at the receptor and have agonist-like effects. In contrast, in our electrophysiological studies in slices we observed that cholesterol reduced GABA receptor-mediated synaptic activity. Furthermore, hormone therapy has been evaluated in a transgenic mouse model of HD (i.e. R6/1 model) (Hannan et al., 2011). The authors showed that hormone treatment had no effects on motor (rotarod) and cognitive deficits (Y-maze) but induced a hypo-locomotion in both wt and HD mice. On the contrary, we found that cholesterol supplementation had no effects on motor deficits but rescued cognitive and synaptic related deficits and partially improved global activity in HD-R6/2 mice. In addition, an increased production of steroids should lead to an increase of the body weight. However, we found that all the groups of R6/2 mice used in this work lose body weight over time.

Ideally, authors should include a 5th (wt) and 6th (R6/2) groups of mice for the rescue (electrophysiology and behavior) studies: cholesterol nanoparticles without the g7 carrier (no brain entry of cholesterol).

At the beginning of this project, a preliminary trial was initially performed in wt and R6/2 mice treated with control NPs loaded with cholesterol (without g7, i.e. not able to cross the BBB). No changes were found in terms of behavioral tasks and molecular signature. Therefore, we decided to not include these groups in subsequent trials with g7-NPs-Chol. This is now indicated in the method section (pag. 13 in the revised text).

Referee #3 (Remarks):

This manuscript is built on previous interesting results from the authors showing deficits in

cholesterol metabolism in the neurodegenerative Huntington disease (HD). Here they have developed glucopeptide modified nanoparticles to deliver cholesterol into a commonly used mouse model for the disease, the R6/2 mouse. They convincingly show that these particles cross the BBB and that the product is being taken up by both glial and neuronal cells. They show that this treatment has a positive effect on synaptic function, general activity as well as cognitive function in the mice.

This is a very well and carefully performed study that is written in a clear and logical fashion. The results are well presented, and the interpretation and discussion of the results are well balanced. The data from this study holds promise for new treatment possibilities for HD although much research is still needed in order to further develop this therapy and improve its efficacy.

We thank the referee for his/her comments.

2nd Editorial Decision

07 October 2015

Thank you for the submission of your revised manuscript to EMBO Molecular Medicine. We have now received the enclosed reports from the referees that were asked to re-assess it. As you will see even though referee 1 remains hesitant regarding the appropriateness of the mouse model used, in light of referee 2 evaluation and following editorial further discussions, I am pleased to inform you that we will be able to accept your manuscript pending the following final amendments:

1) We agree with referee 2 that addition of the non-presented data would be valuable for the readers and we would like to encourage you to include and discuss it

I look forward to reading a new revised version of your manuscript as soon as possible.

***** Reviewer's comments *****

Referee #1 (Remarks):

The authors did not address any of my comments. In particular, I think that it was reasonable to ask the authors to reproduce some key experiments showing an amelioration of the synaptic and cognitive functions (electrophysiology, behavior) in a full-length model of HD (e.g. zQ175). Indeed, 1) the phenotypes observed with the R6/2 mouse model are weak and 2) there is conflicting results in the literature on the levels of cholesterol in HD and the use of cholesterol supply as a therapeutic approach in HD. Thus, the R6/2 model is not sufficient to warrant publication of a therapeutic strategy in EMBO Molecular Medicine.

Referee #2 (Remarks):

I am overall happy with (most of) the answers. Although I would have liked to have, at least, some experimental data that could assess the effect of peripheral cholesterol particles on steroid production, my major concern (whether the central effects are due to a central impact or peripheral) is now addressed by the authors description of an experiment with cholesterol nano particles unable to cross the blood brain barrier. Concerning this point however, authors argue that they did this experiment at the beginning of the project, not observing any changes in terms of behavioral tasks and molecular signature and had consequently omitted these data in the current version. Because of the importance of this result I believe authors and readers would benefit by including these results, perhaps as supplementary material, and describe this result .

1) referee 2 and the editor have requested the inclusion of unpublished data

In this version we now include the data about the initial trial performed in WT and HD mice treated with unmodified control NPs loaded with cholesterol (C-NPs-Chol), i.e. not able to cross the BBB. In particular, we show the absence of a rescue in brain *bdnf* levels and of behavioral phenotype in R6/2 mice treated with C-NPs loaded of cholesterol compared to controls. Please see the new Appendix Figure S11.